# Environment-induced epigenetic reprogramming in genomic regulatory elements in smoking mothers and their children

Tobias Bauer[1,†], Saskia Trump[2,†], Naveed Ishaque[1,3,†], Loreen Thürmann[2,†], Lei Gu[1,†,‡], Mario Bauer[2,†], Matthias Bieg[1,3], Zuguang Gu[1,3], Dieter Weichenhan[4], Jan-Philipp Mallm[5], Stefan Röder[2], Gunda Herberth[2], Eiko Takada[6], Oliver Mücke[4], Marcus Winter[2], Kristin M Junge[2], Konrad Grützmann[2], Ulrike Rolle-Kampczyk[7], Qi Wang[1], Christian Lawerenz[1], Michael Borte[8], Tobias Polte[2,9], Matthias Schlesner[1], Michaela Schanne[10], Stefan Wiemann[10], Christina Geörg[11], Hendrik G Stunnenberg[12], Christoph Plass[4], Karsten Rippe[5], Junichiro Mizuguchi[6], Carl Herrmann[1,13,†], Roland Eils[1,3,13,14,*,†] & Irina Lehmann[2,**,†]

## Abstract

Epigenetic mechanisms have emerged as links between prenatal environmental exposure and disease risk later in life. Here, we studied epigenetic changes associated with maternal smoking at base pair resolution by mapping DNA methylation, histone modifications, and transcription in expectant mothers and their newborn children. We found extensive global differential methylation and carefully evaluated these changes to separate environment associated from genotype-related DNA methylation changes. Differential methylation is enriched in enhancer elements and targets in particular "commuting" enhancers having multiple, regulatory interactions with distal genes. Longitudinal whole-genome bisulfite sequencing revealed that DNA methylation changes associated with maternal smoking persist over years of life. Particularly in children prenatal environmental exposure leads to chromatin transitions into a hyperactive state. Combined DNA methylation, histone modification, and gene expression analyses indicate that differential methylation in enhancer regions is more often functionally translated than methylation changes in promoters or non-regulatory elements. Finally, we show that epigenetic deregulation of a commuting enhancer targeting c-Jun N-terminal kinase 2 (JNK2) is linked to impaired lung function in early childhood.

**Keywords** environment; epigenetics; WGBS; histone modifications; enhancer deregulation
**Subject Categories** Chromatin, Epigenetics, Genomics & Functional Genomics; Genome-Scale & Integrative Biology; Systems Medicine
**Mol Syst Biol.** (2016) 12: 861

## Introduction

Altered epigenetic patterns represent an attractive explanation for understanding the phenotypic changes associated with environmental exposure. By affecting DNA methylation, post-translational histone modification, or non-coding RNA (ncRNA) signaling, environmental factors may cause persistent perturbations of regulatory

1   Division of Theoretical Bioinformatics, German Cancer Research Center (DKFZ), Heidelberg, Germany
2   Department of Environmental Immunology, Helmholtz Centre for Environmental Research Leipzig - UFZ, Leipzig, Germany
3   Heidelberg Center for Personalized Oncology, DKFZ-HIPO, DKFZ, Heidelberg, Germany
4   Division of Epigenomics and Cancer Risk Factors, German Cancer Research Center (DKFZ), Heidelberg, Germany
5   Research Group Genome Organization & Function, German Cancer Research Center (DKFZ) and Bioquant, Heidelberg, Germany
6   Department of Immunology, Tokyo Medical University, Tokyo, Japan
7   Department Metabolomics, Helmholtz Centre for Environmental Research Leipzig - UFZ, Leipzig, Germany
8   Municipal Hospital "St Georg" Children's Hospital, Leipzig, Germany
9   Department of Dermatology, Venerology and Allerology, Leipzig University Medical Center, Leipzig, Germany
10  Genomics and Proteomics Core Facility, German Cancer Research Center (DKFZ), Heidelberg, Germany
11  Sample Processing Lab, National Center for Tumor Disease and German Cancer Research Center (DKFZ), Heidelberg, Germany
12  Department of Molecular Biology, Faculty of Science, Radboud University, Nijmegen, The Netherlands
13  Institute of Pharmacy and Molecular Biotechnology and Bioquant Center, University of Heidelberg, Heidelberg, Germany
14  Translational Lung Research Center Heidelberg (TLRC), German Center for Lung Research (DZL), University of Heidelberg, Heidelberg, Germany
    *Corresponding author. E-mail: r.eils@dkfz-heidelberg.de
    **Corresponding author. E-mail: irina.lehmann@ufz.de
    †These authors contributed equally to this work
    ‡Present address: Division of Newborn Medicine, Children's Hospital Boston and Department of Cell Biology, Harvard Medical School, Boston, USA

pathways and thus induce an altered susceptibility for disease (Aguilera *et al*, 2010). Many lines of evidence indicate that early life and in particular the prenatal period represent a window of high vulnerability to environmental impacts with consequences for disease risk later in children's life. The subsequent manifestation of a disease may occur with long latency periods as shown in the Dutch Hunger Winter study. This study revealed that starvation during pregnancy increased the risk for several diseases later in children's life including type II diabetes, cardiovascular diseases, or decreased cognitive function (Brown *et al*, 1995; Susser *et al*, 1996; Roseboom *et al*, 2001; Painter *et al*, 2005; Veenendaal *et al*, 2013). Experimental models suggest that the *in utero* nutritional environment resulting from starvation induces epigenetic modifications including altered DNA methylation (Radford *et al*, 2014) and the generation of small RNAs (Rechavi *et al*, 2014) that are inherited in the next generation.

One of the most common hazardous prenatal exposures is maternal smoking. Prenatal exposure to tobacco smoke was described as a risk factor for a multitude of different diseases in the child, including lung diseases, obesity, and cancer (Hemminki & Chen, 2006; Oken *et al*, 2008; Neuman *et al*, 2012). Several studies have focused on DNA methylation in cord blood to elucidate the influence of smoking and other prenatal exposures on the newborn's epigenome by analyzing global DNA methylation changes (i.e., in repetitive elements) or methylation of a limited number of preselected CpG probes (i.e., 27k methylation or 450k arrays) (Breton *et al*, 2009; Joubert *et al*, 2012; Murphy *et al*, 2012; Markunas *et al*, 2014; Kupers *et al*, 2015; Richmond *et al*, 2015). From these earlier epidemiological studies, information on global and site-specific methylation changes is available, but the insights derived from those studies remain very limited. Importantly, these earlier investigations, being based on 450k methylation array data, covered only a small fraction of the genome and, for example, lack information on enhancer regulatory elements located outside of promoters.

Although changes of epigenetic modifications due to environmental cues early in life may persist over time, genome-wide data for studying longitudinal stability of epigenetic patterns in humans are still missing. Earlier studies have shown evidence for long-term stability of a limited number of methylation loci. However, to what extend this is a global trend, or only limited to some methylation loci is unclear, given the sparse coverage of the probes from the methylation array. In this study, we address the following questions: When and where in the genome are epigenetic marks set by environmental factors? What is the contribution of the genetic sequence variation to changes in DNA methylation? Do those changes that are associated with maternal smoking persist over years or do they appear only transiently? Furthermore, do DNA methylation changes contribute to early programming for disease?

To address these questions, we performed a comprehensive epigenetic characterization within the LINA mother–child birth cohort (Herberth *et al*, 2006, 2011) to dissect the link between environmental exposure and epigenetic signals. We first studied the DNA methylome at single base pair resolution in both children and their mothers around time of birth and until 4 years after birth. To decipher the regulatory role of DNA methylation changes associated with smoking, we performed histone modification ChIP-seq of four histone modification marks to segment the genome into distinct

regulatory elements and linked the environmentally associated differential DNA methylation to transcription as measured by RNA sequencing. Finally, we show that DNA methylation changes in conjunction with histone modifications are related to disease development later in children's life.

## Results

### Maternal smoking is associated with genome-wide DNA methylation changes that are different between mothers and their children

A variety of studies has investigated the impact of maternal smoking on epigenetic changes in newborn children and assessed the stability of such epigenetic marks over time. Note that these studies focused on DNA methylation changes disregarding equally important changes on the chromatin level (Joubert *et al*, 2012; Kupers *et al*, 2015; Lee *et al*, 2015; Richmond *et al*, 2015). Further, these studies were performed on the basis of DNA methylation arrays covering as few as 5% of the entire set of CpG dinucleotides in the genome located primarily in promoter regions. To overcome the limitations of presently widely used DNA array methylation arrays offering only a narrow view on the DNA methylome, we here followed a radically different discovery and validation strategy (Fig 1). First, we set out to perform whole-genome bisulfite sequencing (WGBS) in a set of mothers alongside with their children and we went further to seek functional support for our findings. For that, we performed detailed analysis of chromatin configuration changes encompassing DNA methylation changes and further studied genome-wide changes in gene expression by RNA sequencing as a functional readout of the concerted action of DNA methylation and chromatin configuration changes. Finally, we set out to validate selected findings in the entire discovery cohort and in an independent replication cohort (Fig 1).

To study the impact of maternal smoking on DNA methylation in both mothers and children, we performed WGBS of whole blood samples from 32 individuals (maternal blood at 36th week of gestation for eight smoking mothers and eight non-smoking mothers and cord blood from their respective child; Table EV1; Fig EV1) at up to three different time points (for three children from smoking and three from non-smoking mothers at year one and four, and their respective mothers at year one; Table EV1; Fig EV1; for overview of discovery/validation strategy, see Fig 1).

Heterogeneity of whole blood samples may be a possible confounder in our analysis. Therefore, we excluded the possibility that differences in cell type composition in the smoking exposed/non-exposed groups would give rise to DNA methylation changes by assessing promoter methylation levels from seven lineage markers reflecting the blood cell type composition in each sample (see Appendix Supplementary Methods). The analysis showed that the variation in cellular composition in response to tobacco smoke exposure was in the range of 1–7% for main blood cell types in mothers and children with significant reduced granulocyte and increased B-cell numbers in newborn children from smoking mothers (Table EV2). To exclude differentially methylated regions (DMRs) that were solely caused by differences in cellular blood composition between exposed and non-exposed samples, we used a

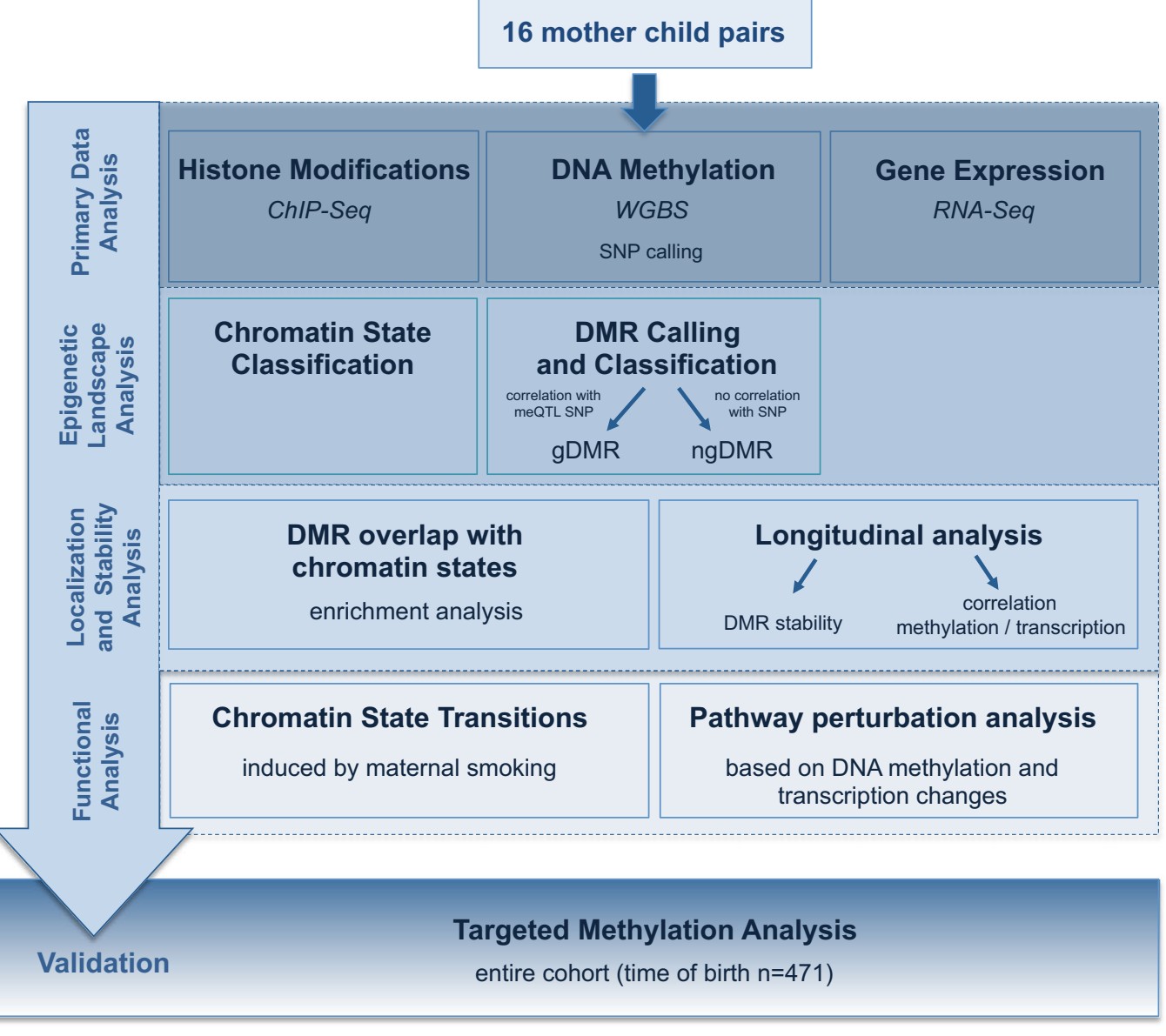

**Figure 1. Discovery and validation strategy.**
To overcome the limitations of presently widely used DNA array methylation arrays offering only a narrow view on the DNA methylome, we followed a different discovery and validation strategy. First, we performed whole-genome bisulfite sequencing (WGBS) in a set of mothers alongside with their children. WGBS was carried out in 16 mother–child pairs, a rather small sample number compared to the typical sizes of epigenetic studies performed on DNA methylation arrays. However, we compensated this potential drawback by a subsequent comprehensive functional validation. We performed a detailed analysis of chromatin configuration and DNA methylation changes—including longitudinal stability analyses—and further studied genome-wide gene expression by RNA sequencing as a functional readout of the concerted action of the observed epigenetic perturbations. In a targeted analysis, we finally validate selected findings in the entire discovery cohort.

threshold of 10% for DMR calling (Fig EV2, Appendix Supplementary Methods).

Previous studies assessing the impact of smoking on differential DNA methylation had reported changes for single CpGs in the range of 1–25% (Zeilinger *et al*, 2013), indicating that our requirement of a 10% change represents a conservative threshold. To ensure consistently different methylation for all samples between groups, we excluded DMRs by a moderated t-statistics ($P > 0.1$) and permutation analysis. We thus identified 9743 and 8409 significant

($P < 0.1$, $\Delta_{\text{Methylation}} > 0.1$) DMRs in mothers and children, respectively, in the comparison of the smoking and the non-smoking individuals at time of birth (Fig 2, Table EV3). Using a random shuffling procedure, we determined that the median false discovery rate (FDR) level was 12.4% for children and 11.2% for mothers (see Appendix Supplementary Methods). Note that less than 5% of these DMRs are covered by CpG probes from the 450k platform. We finally conducted a FDR analysis based on permutation analysis of DMRs with subsequent filtering with a range of thresholds for

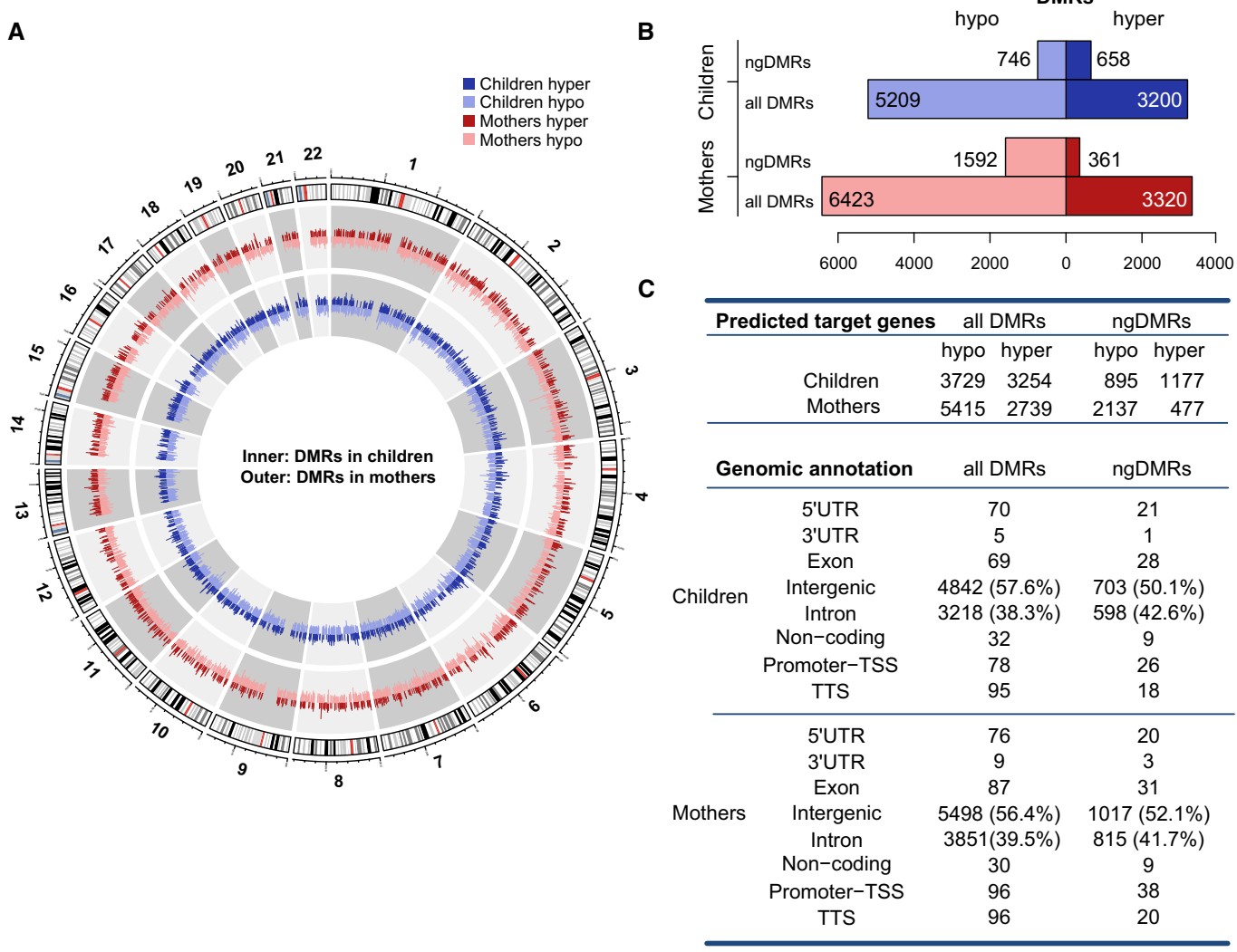

**Figure 2.  Mothers and children harbor a large number of differentially methylated regions between smokers and non-smokers at time of birth.**

A  Circular representation of DNA methylation levels for mothers (outer circle) and children (inner circle). The height of each bar indicates the methylation change between the smoking and non-smoking group (dark hue: hypermethylation, light hue: hypomethylation).

B  Bar plots represent the number of hypo- versus hypermethylated DMRs for all DMRs and ngDMRs in children and mothers.

C  Number of genes predicted to interact with DMRs and ngDMRs. Annotation of allDMRs/ngDMRs according to genomic categories (lower panel).

$\Delta_{\mathrm{Methylation}}$ (1–25%). The obtained results implied that the 10% cutoff offers an optimal balance between high sensitivity and medium specificity, which are required when comparing groups of healthy individuals (see Appendix Supplementary Methods, Fig EV2).

Beyond environmental factors, the individual genotype also effects DNA methylation. A single nucleotide polymorphism (SNP) may destroy the CpG context and thus directly induce differential DNA methylation *in cis* by reducing the methylation to 0 (0.5) in case of a homozygous (heterozygous) CpG-destroying SNP. Further, SNPs may induce differential methylation by either creating or disrupting a transcription factor binding site that may affect the level of DNA methylation (Gutierrez-Arcelus *et al*, 2013) and may alter histone modifications (Kasowski *et al*, 2013; Kilpinen *et al*, 2013; McVicker *et al*, 2013). Given our sample size, we cannot exclude

the fact that in some cases, a differential methylation between the two groups is due to genotype rather than environmental effects.

To address this issue, we performed SNP calling for each individual bisulfite sequencing sample using the bisulfite-conversion-aware SNP caller Bis-SNPs (Liu *et al*, 2012). To distinguish between environmental and genotype-associated differential DNA methylation, we categorized the DMRs into two groups. Whenever we found a SNP in the neighborhood (± 5 kb) of a DMR, for which the genotype was statistically significantly correlated (at 10% FDR, see Appendix Supplementary Methods) with methylation of this DMR, we call this SNP a meQTL-SNP (methylation quantitative trait locus) for this DMR (see Fig EV3). The corresponding DMR is then termed a potentially genetically influenced DMR and abbreviated as gDMR in the following. All DMRs that are not associated with any meQTL-SNP in their neighborhood are presumably not influenced by the

genotype and thus termed non-genetically influenced DMR (ngDMR). Even with this highly stringent definition, we retain about 17–20% (1404/8409 in children and 1953/9743 in mothers) of all DMRs as ngDMRs that do not have an apparent correlation with the underlying genetic sequence variation. In summary, we determined a set of 1404 (1953) DMRs in children (mothers) at time of birth, which are not related to genotype effects and are likely enriched in regions for which the environmental cue drives methylation changes.

## DNA methylation changes due to maternal smoking are stably maintained over years of life

It has been discussed controversially whether DNA methylation remains stable over longer periods of time or whether there is a general loss of methylation associated with aging (Heyn *et al*, 2012; Raddatz *et al*, 2013). Previous studies based on 450k data have shown some examples of CpGs whose methylation levels are stably maintained over years (Guida *et al*, 2015; Lee *et al*, 2015; Richmond *et al*, 2015). For example, *Richmond et al* described 3 CpGs showing a stable methylation difference related to early tobacco exposure persisting until the age of 17. Our population-based cohort offers the unique opportunity to study longitudinal DNA methylation stability in one and the same individual by comparing two different time points separated by several years for all CpGs in the genome. We performed WGBS sequencing for six mothers and their children (three smoking and three non-smoking; Fig EV1, Table EV1 for sample overview) 1 year after birth and for the same six children 4 years after birth (for a prototypical example see Fig 3A).

To assess the stability of methylation over time within one individual compared to other individuals, we performed hierarchical clustering of all CpGs located within ngDMRs with coverage > 10 ($n = 4682$ in children, $n = 7857$ in mothers). Interestingly, the methylomes from the same individual at different years (children: time of birth, 1 and 4 years thereafter; mothers: time of birth and 1 year thereafter) clustered perfectly (Fig EV4A and B) even though we had removed all genotype-associated gDMRs prior to clustering. This suggests that a large fraction of specific DNA methylation sites is stably maintained over at least 4 years in children and over 1 year in mothers.

Next, we conducted a genome-wide analysis to assess whether DMRs called at time of birth would remain differentially methylated 1 year after birth. Using the hyper- and hypomethylated regions determined at time of birth, we assessed whether these regions remained in their respective methylation state. In 82% (90.4%) of all ngDMRs in children (mothers), there was still a hypermethylation and hypomethylation 1 year after birth corresponding the methylation change observed at time of birth (qualitative stability, Fig 3B, middle panel). Even with the additional constrain that the maximum decrease/increase was ≤ 5% of the methylation level at time of birth, ngDMRs were retained in 73.5% in mothers and in 59.7% in children (quantitative stability, Fig 3B, middle panel). In children, differential DNA methylation remains at the same level 4 years after birth. Note that the level of stability only increased by 2.6% in mothers and 10.7% in children, respectively, comparing qualitatively stable ngDMRs with stable gDMRs (Fig 3B lower panel). Thus, the contribution of the genotype to longitudinal stability of differential DNA methylation was considerably low. In summary,

maternal smoking is associated with differential DNA methylation that is persistent in mothers and children up to 4 years after birth.

## Tobacco smoke-associated DNA methylation changes correlate with hypervariable chromatin

To evaluate whether differential DNA methylation correlated with active or repressive regulatory elements in the genome, we mapped histone modifications by ChIP-seq for six mothers and their children for which we already performed WGBS, around year four after birth (Table EV1). We selected four histone marks to delineate active/poised elements (H3K4me1 and H3K27ac), separated as enhancers or promoters based on proximity to a gene's TSS, as well as repressive elements (H3K9me3 and H3K27me3; Fig 4A). Bivalent states were defined by co-occurrence of active and repressive marks.

To segment the genome into distinct functional elements, ChromHMM (Ernst & Kellis, 2012), a Hidden Markov model-based approach, was trained to cover all 16 possible combinations of the four histone modifications (Fig 4B) and then applied to segment the genome of each individual. Note that we started with a fine-grained definition of 16 chromatin states, but most of the subsequent analysis is done by collapsing the three active and repressive states into an active and repressive meta-state, respectively. The vast majority (91% in mothers, 90.8% in children; Table EV4) of the genome across all individuals was not covered by any of the four histone marks (background state "void"). The three active/poised enhancer and active promoter states in children or mothers covered about 4.0 and 3.3%, respectively, of the genome, whereas corresponding values for the three repressive states were 1.8 and 2.2%. The remaining part of genomic loci was in a bivalent state and carried both activating and repressive marks simultaneously.

Next, environmentally associated transitions between chromatin states were identified from a comparison of non-smoking and smoking individuals (Fig 4C; Table EV5). For mothers, we observed slightly more transitions to repressive states than to active (29.7% active versus 41.7% repressive transitions). In children, the vast majority (43.9%) of transitions were into active states, whereas transitions into repressive state were much less frequent (17.4%). This suggests that maternal smoking is associated with a hyperactive chromatin state in their children. While genome-wide chromatin state transitions were only statistically significant for transitions into repressed states in mothers and into bivalent states in children and in mothers, all of the transitions overlapping with ngDMRs were statistically significant (Table EV5). These significance levels were much more pronounced within ngDMRs, while all transitions within gDMRs for children reach only borderline significance levels. Together with our observation of a significant enrichment (2.4 × in children and 3.0 × in mothers) of variable chromatin in the ngDMRs, these findings suggest that much of the chromatin dynamics is strongly linked with differential methylation related to maternal smoking.

## Tobacco smoke exposure is associated with epigenetic reprogramming of regulatory genomic elements

Only a minor fraction of DMRs co-localized with regions close to TSS (Fig 2C). Thus, we examined whether the remaining DMRs fall into active or repressive regulatory genomic elements outside of the

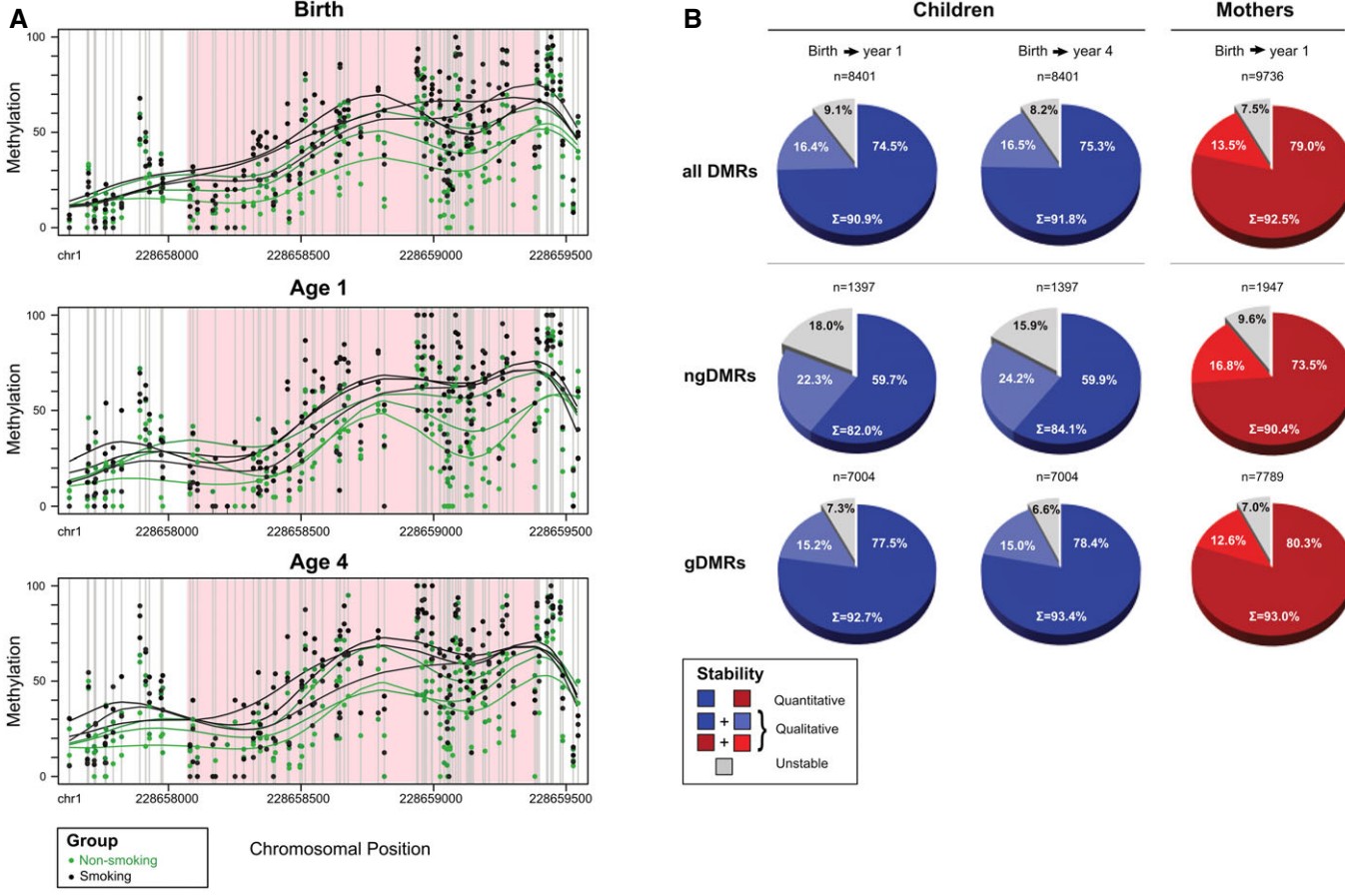

**Figure 3. Differentially methylated regions show high stability over time.**

A  Example of an intergenic ngDMR located 8,965 bp away from the TSS of miR-466-6A. Green dots indicate the raw methylation values for non-smoking samples for all 48 CpGs in the region, while black dots indicate the raw methylation of children from smoking mothers. Lines represent smoothed methylation levels. Methylation differences of 14.5, 13.5, and 12.5%, at time of birth, one year after birth, and four years after birth, respectively, show a strong, quantitative stability of the differential hypermethylation at this locus.

B  Global analysis over all DMRs, ngDMRs, and gDMRs (from top to bottom row) in mothers and children shows stability of methylation using both quantitative criteria (decrease in absolute mean methylation difference between smokers and non-smokers should not exceed 5%) as well as qualitative criteria (direction of differential methylation should remain identical irrespective of the absolute methylation change). As expected, genetically determined gDMRs are more stable than their non-genetically determined counterparts. Still, 90.4% (82–84.1%) of the ngDMRs show longitudinal stability in mothers (children).

TSS. First, we defined active/poised (state 1–3) and repressive (state 12–14) enhancer meta-states (see genome segmentation above and Appendix Supplementary Methods for details). Note that active regulatory elements outside of the TSS were considered to be active/poised enhancers based on their ChIP-seq profile, whereas active regulatory elements in proximity of TSS were considered as active promoters. About twice as many DMRs were located in active regulatory elements outside TSS ("enhancers") than close to TSS ("promoters") both in mothers and in children (Fig 5A and Table EV7). This trend is observed for all DMRs, but also for ngDMRs and gDMRs separately. For repressive regulatory elements, the ratio between DMRs in such elements was approximately 7:1 when comparing regions outside and in proximity of TSS. This stronger enrichment of differential methylation in regulatory regions outside TSS was not a mere consequence of the relatively small genomic size of the TSS region. Rather, by permutation analysis (see Appendix Supplementary Methods for details), we observed a

highly significant ($P = 1.69\text{e-}66$ for mothers, $P = 1.74\text{e-}55$ for children) enrichment of ngDMRs in enhancers both in mothers and in children (Fig 5A and Table EV7). The enrichment of ngDMRs in promoters was considerably lower ($P = 3.37\text{e-}23$ for mothers and $P = 6.26\text{e-}25$ for children). Similarly, ngDMRs in repressive elements showed a highly significant enrichment outside TSS ($P = 9.33\text{e-}16$ for mothers and $P = 1.89\text{e-}18$ for children), but considerably lower significance ($P = 1.56\text{e-}4$ for mothers and $P = 5.07\text{e-}3$ for children) in promoters. These results suggest that environmentally associated differential methylation targets enhancer elements or repressive elements predominantly outside TSS.

Enhancer regions that overlap with DMRs were called differentially methylated enhancers, or DMEs. As described above for gDMRs and ngDMRs, we distinguished between enhancers in which differential methylation correlated with a SNP within a $\pm$ 5-kb window (gDMEs) versus the remaining cases (ngDMEs). The ngDMEs were mainly located in intragenic regions (62.1% in

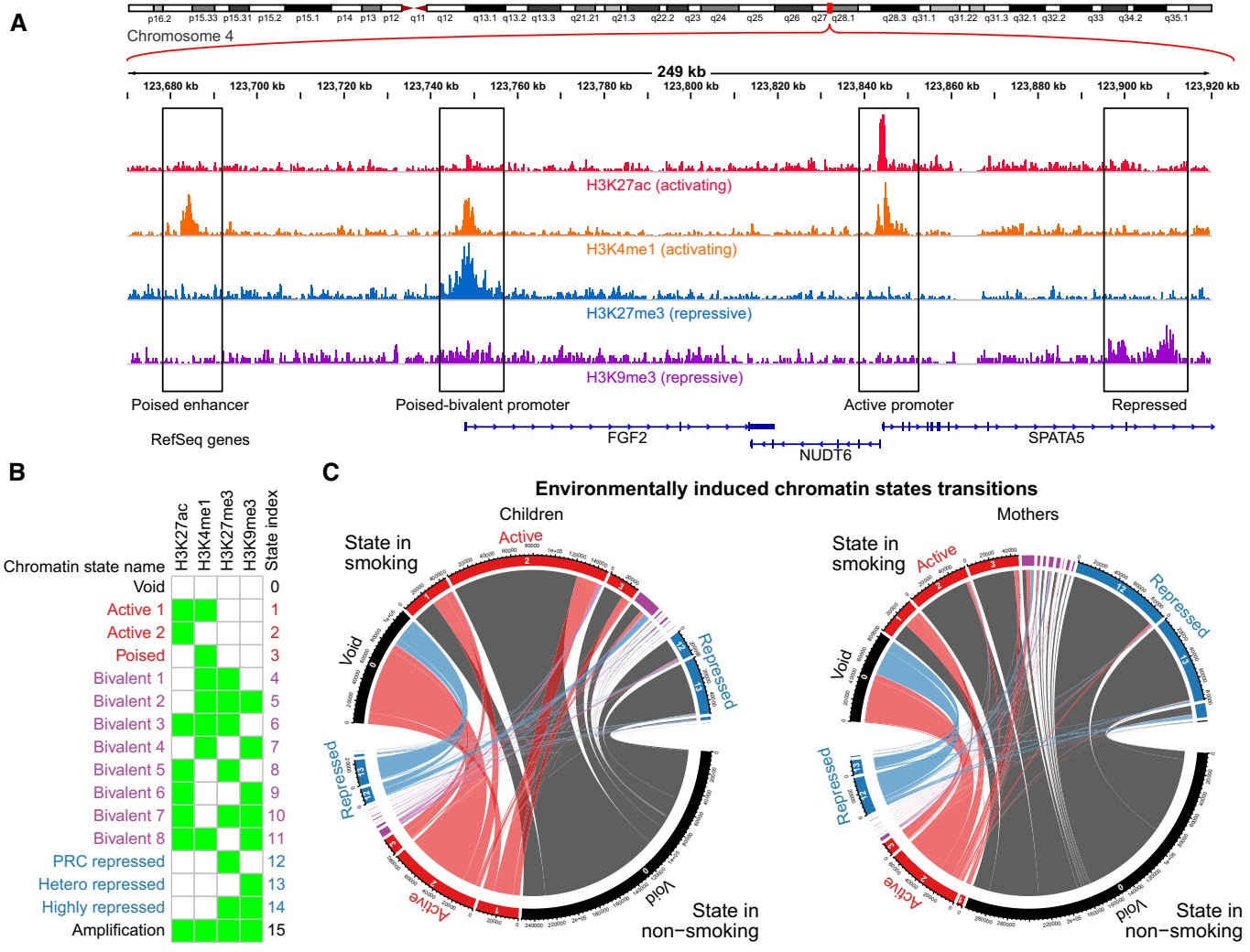

**Figure 4.  Chromatin states built from histone marks show specific patterns of transition between non-smokers and smokers..**

A   Visualization of four histone mark read densities over 249 kb of chromosome 4 representing active marks (H3K27ac and H3K4me1) and repressive marks (H3K27me3 and H3K9me3). This region illustrates examples of (boxed from left to right): a poised enhancer upstream of FGF2 (exhibiting H3K4me1, but lacking the active signature of H3K27ac); a bivalent promoter of FGF2 (with co-occurring active H3K4me1 and repressive H3K27ac at the promoter); active promoters marks around the TSS of NUDT6 and SPATA5 (displaying the 2 active marks of H3K27ac and H3K4me1); and repressive marks in the gene body of SPATA5 (H3K9me3), which we observe in the intron of some expressed genes.

B   Chromatin state model derived from all samples representing all possible 16 states. The states were reorganized and named according to biologically relevant chromatin states representing: lack of signal (state 0, "Void"); active states (states 1–3, in red); bivalent states, showing a combination of both active and repressive marks, organized by the proportion of the genome that they represent on average (states 4–11, in purple); repressed states (states 12–14, in blue); and a state representing co-occurrence of all marks, most likely representing genomic amplifications (state 15, "Amplification").

C   Chord diagram of chromatin state transitions in unstable regions of chromatin. The plot shows the transitions from non-smoking to smoking for children and mothers, where the size of the outer segments represents the amount of chromatin undergoing transitions from one state to another and the width of the ribbons represents the amount of a single state transiting to another chromatin state. The scale of segment size and ribbon width are comparable between mothers and children. The outer segments and ribbons are colored according to transition to/from void, active, bivalent, repressed, and amplified as black, red, purple, blue, and black, respectively. It can be observed that in children, there is a net overall transition to a more hyperactive chromatin state (i.e., gain of twice as much active chromatin compared to repressed), which is not observed in the mothers who have a more balanced gain of both active and repressive states.

mothers, 59.5% in children; Fig 5A). Comparable numbers were found for gDMEs. To identify DME target genes, we used various datasets from genomic interactions based on ChIA-PET assays or predicted DHS–promoter interactions (see Appendix Supplementary Methods for details on used interaction data). Interestingly, 93.2% (88%) of intragenic ngDMEs in children (mothers) are annotated to target at least one gene outside the host gene in (Table EV8). We

call these elements "commuting" enhancers, as they reside in one gene but act on various distal genes (Fig 5B). About a third of those commuting ngDMEs even did not interact with their host gene and were termed "exclusive commuting enhancers" (Fig 5B). DMEs on average covered larger genomic regions than non-DMEs (Fig EV5A) and showed significantly more interactions with predicted target genes than any other enhancers for intragenic elements

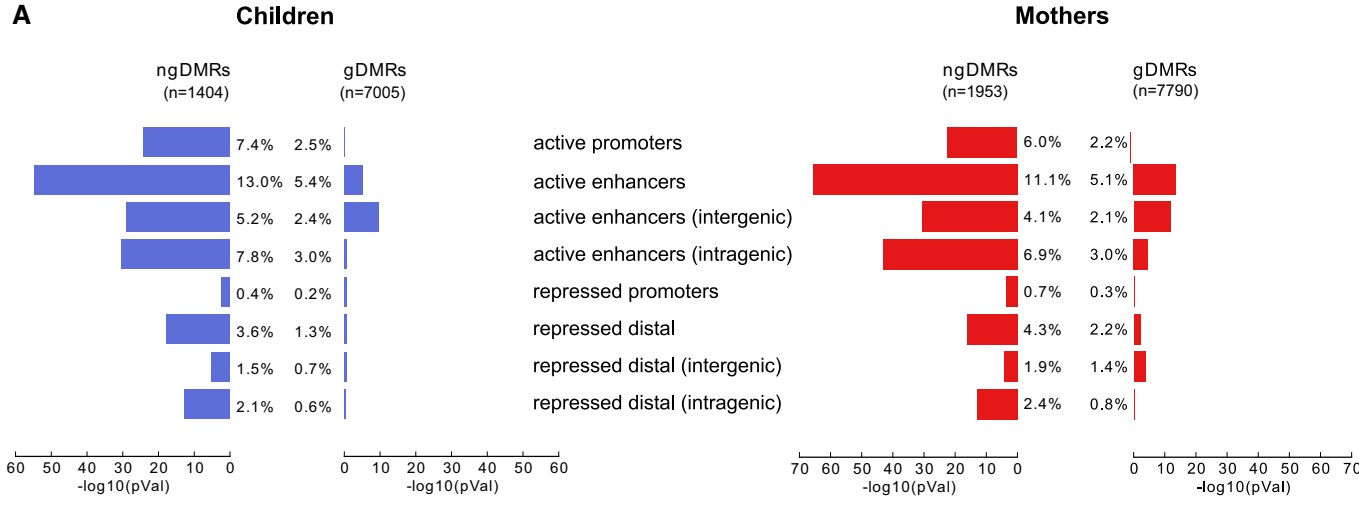

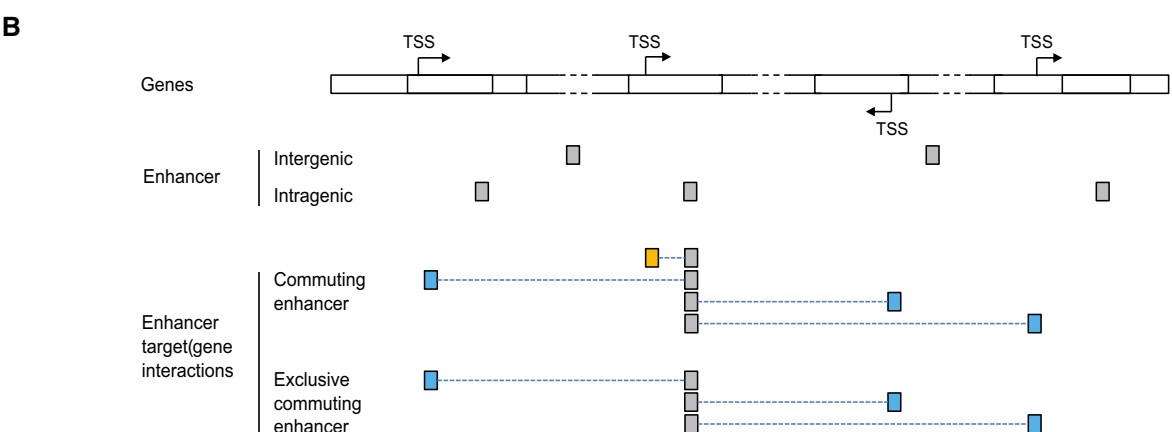

**Figure 5.  Differential methylation mostly targets distal intragenic regulatory regions.**

A    Enrichment of DMRs in annotated chromatin elements for children (left) and mothers (right). The bar plot shows the −log10(*P*-value) of the enrichment for ngDMRs (left bars) and gDMRs (right bars). The percentages indicate the proportion of g/ngDMRs that overlap with chromatin elements of the category.

B    Definitions of various enhancer classes: if at least one gene outside the host gene is regulated, we call it a "commuting" enhancer, as it resides in one gene but act on distal gene(s). When commuting enhancers do not interact with their host gene, we call them "exclusive commuting enhancers".

($P$ = 5.6e-6/1.2e-4 in children/mothers, Mann–Whitney test) and marginally significant more for intergenic elements ($P$ = 0.015/0.037 in children/mothers; Fig EV5B). The results were comparable when restricting to the subset of enhancers overlapping ngDMRs (ngDMEs, Fig EV5A and B, right column).

**Transcriptional response of epigenetic reprogramming of regulatory genomic elements**

We next set out to understand whether environmentally associated epigenetic changes would be reflected by differential expression between the smoking and non-smoking groups in mothers and their children. Therefore, we performed RNA-seq analysis for 15 mothers and 12 children at time of birth, for which we already had WGBS data (Fig EV1 for sample overview). When calling differentially expressed genes genome wide, we identified only a very small number (< 10 for children, data not shown) of differentially

expressed genes after correction for multiple testing (10 % FDR, BH correction). To better capture subtle changes of gene expression when comparing groups of healthy individuals, we sought to first understand which pathways are enriched for genes targeted by differentially methylated regions and then to focus our analysis on genes in those pathways. Given the enrichment of DMRs in active enhancers and active promoter regions described in the previous section, we decided to focus on the subset of DMRs, which intersect these active chromatin elements.

To identify target genes of DMRs, we used various datasets from genomic interactions based on ChIA-PET assays or predicted DHS–promoter interactions (see Appendix Supplementary Methods for details and Table EV3 for assigned genes). Among the pathways significantly enriched in both mothers and children, we identified the WNT signaling pathway (Table EV6). We found 62 genes belonging to the WNT signaling pathway, which are targets of DMRs in smoking mothers and 48 of such genes in their newborn

children (Table EV6, Fig EV6). Aberrant WNT signaling has been reported to be involved in the airway inflammatory response in healthy smokers and smokers with chronic obstructive pulmonary disease (COPD) (Wang *et al*, 2011) and has also been implicated in a variety of tumors, including lung cancer (Ying & Tao, 2009). About half of the pathways enriched by genes targeted by regions that are differentially methylated between the smoking and non-smoking groups overlapped between children and mothers (Table EV6A). In contrast, the vast majority of pathways enriched for target genes of ngDMRs were different for mothers and their children (Table EV6B) indicating a clear difference in environmental modulation of DNA methylation patterns in mothers and children.

About a quarter ($n = 22$) of all pathways enriched for differential DNA methylation displayed differential expression for the genes in those pathways for mothers or children (Fig EV7A). Among all pathways enriched for DMRs, the WNT pathway in children, but not in mothers, was the pathway with the highest level of significance of differential expression (Fig EV7A). Accordingly, differential expression of genes in the WNT signaling pathway resulted in a clear separation between tobacco smoke-exposed/non-exposed samples (Fig EV7B).

More generally, we explored the functional importance of genomic regulatory elements by correlating the DNA methylation level of such elements with transcription of their predicted target genes. In general, correlation between promoter DNA methylation and gene expression was poor and typically only 5–10% of DMRs correlated with gene expression, similar to the observation in a study of chronic lymphocytic leukemia patients (Kulis *et al*, 2012). Within our dataset, also a relatively weak correlation (6–11%) between DNA methylation and transcription in regions void of any chromatin mark in both mothers and children was observed (Fig EV8A). However, we found a high number of predicted target genes whose expression correlated with DNA methylation in regulatory elements. Almost 30% of all differentially methylated repressive regulatory elements showed a positive correlation with expression of target genes in mothers and 15% in newborn children (Fig EV8B). Note that the same tendency is observed when restricting our analysis to ngDMRs (Fig EV8C). On the other hand, 20% of all differentially methylated enhancers were negatively correlated with expression of target genes in mothers and 8% in newborn children (Fig EV8C). We compared these numbers with the proportion of significant correlations for DMRs within non-annotated chromatin regions ("void state"), which we considered as background levels, and found that these were indeed much lower. Interestingly, the degree of correlation was much lower at promoters and just above background ("void state", Fig EV8B and C). This is in line with recent observations that enhancer methylation is a better predictor of gene expression than promoter methylation (Aran *et al*, 2013; Blattler *et al*, 2014). For children, we observe fewer significant correlations between methylation in any of these categories, likely reflecting the fact that epigenetic changes in newborns at time of birth translate on the transcriptional status only later in life. Repeating the correlation analysis 1 year after birth for children largely increased the degree of correlation compared to time of birth (Fig EV8A), suggesting that environmentally associated DNA methylation in children was already present at time of birth with increasing impact on transcription in target genes later in life. Taken together, our integrated analysis of genome-wide DNA methylation, histone modification, and RNA expression identifies enhancers and repressive elements outside TSS, whose differential methylation displays a pronounced correlation with the expression of a large proportion of target genes.

## Validation of tobacco smoke exposure-related differential methylation in the entire cohort

Based on our analysis of epigenetically deregulated enhancers, we selected one ngDMR and one gDMR that showed interactions with more than one target gene (commuting enhancer) to confirm the differential DNA methylation observed in the 8 versus 8 comparison in the entire LINA cohort ($n = 471$, Fig EV1) by MassARRAY-based targeted DNA methylation analysis. As a representative example of a ngDMR in a commuting enhancer region, we chose the ngDMR with highest significance level when comparing children from smoking to non-smoking mothers, which was located in an intron of TMEM241 (transmembrane protein 241, Fig 6A). The hypermethylated ngDMR identified in the cord blood was confirmed in the larger sample set. Concomitant with an increase in maternal urine cotinine concentrations, DNA methylation levels increase in this region (Fig 6B). Comparison between WGBS and MassARRAY suggests that the relatively small effects size observed in the targeted validation analysis is related to the weaker spread in methylation values obtained by MassARRAY (Fig 6C).

For the validation of a genetically influenced gDMR, we selected a commuting enhancer involved in WNT signaling and early inflammatory response. This commuting enhancer located in the intron of GFPT2 (glutamine-fructose-6-phosphate transaminase 2) was previously annotated by ChIA-PET (Li *et al*, 2012) as an intronic enhancer interacting with a large number of target genes including c-Jun N-terminal kinase 2 (JNK2; also known as MAPK9) (Fig 6D). Further inspection of SNPs in a ± 5-kb window of the "JNK2 commuting enhancer" revealed that the DNA methylation level was correlated with SNP rs55901738 located within this DMR (Fig EV9A). This C>A polymorphism is interesting since it does not occur in a CpG context, which would result in direct loss of DNA methylation. Rather, this polymorphism is predicted to create a new binding site for the transcription factor Oct4. According to our RNA sequence data, Oct4 is expressed in both children and mothers in our cohort. Since the presence of an Oct4 binding site was related to hypomethylation (Zimmerman *et al*, 2013), the presence of such a binding site may contribute to the decrease of DNA methylation in individuals with the C>A genotype. In both the LINA discovery cohort (Fig 6E) and the LISA validation cohort (Fig EV9B), DNA methylation of the JNK2 commuting enhancer correlates with the genotype. Notably, the majority of children of smoking mothers in the LINA discovery cohort and of tobacco smoke-exposed children in the LISA validation cohort display a C/A or A/A genotype in this position giving rise to an overall reduced methylation of the JNK2 commuting enhancer in smoking individuals (significant for maternal cotinine level > 350 μg/g creatinine and for children's urine cotinine levels > 40 μg/g creatinine; Table EV9A/B).

In cord blood, a decrease of DNA methylation in the JNK2 enhancer was observed that was related to maternal urine cotinine levels (Fig 6F). This relation points to a combined effect with the genotype predisposing for an environmentally triggered epigenetic deregulation of JNK2. This hypothesis is corroborated by our

**A    ngDMR**

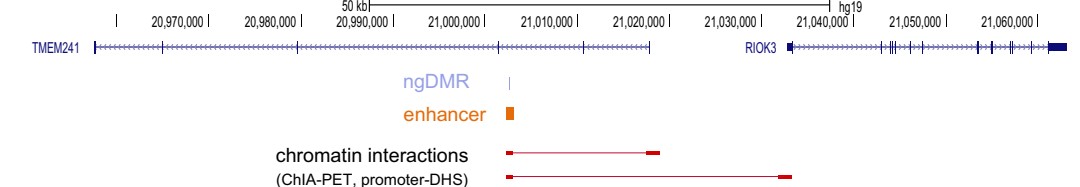

**B**

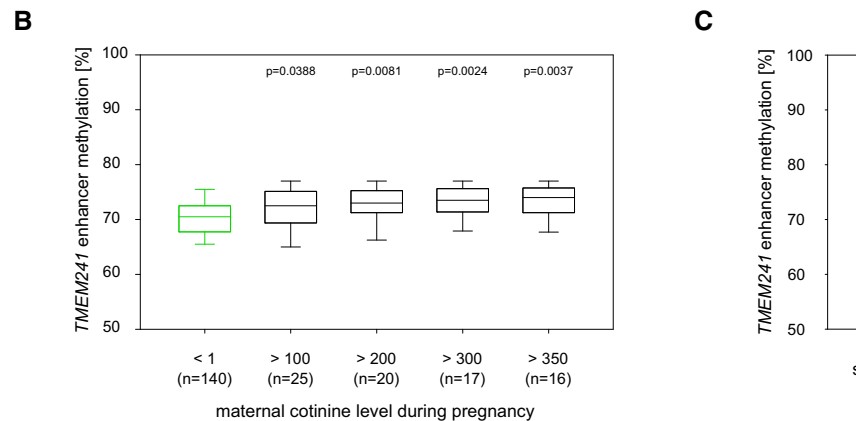

**C**

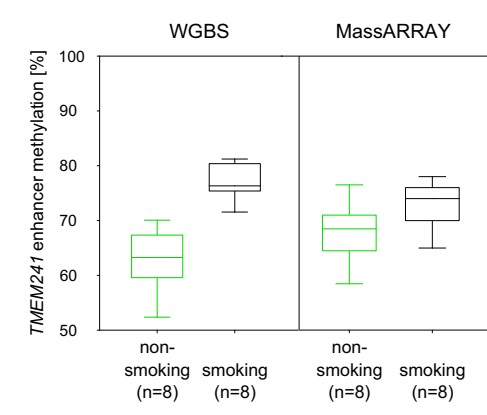

**D    gDMR**

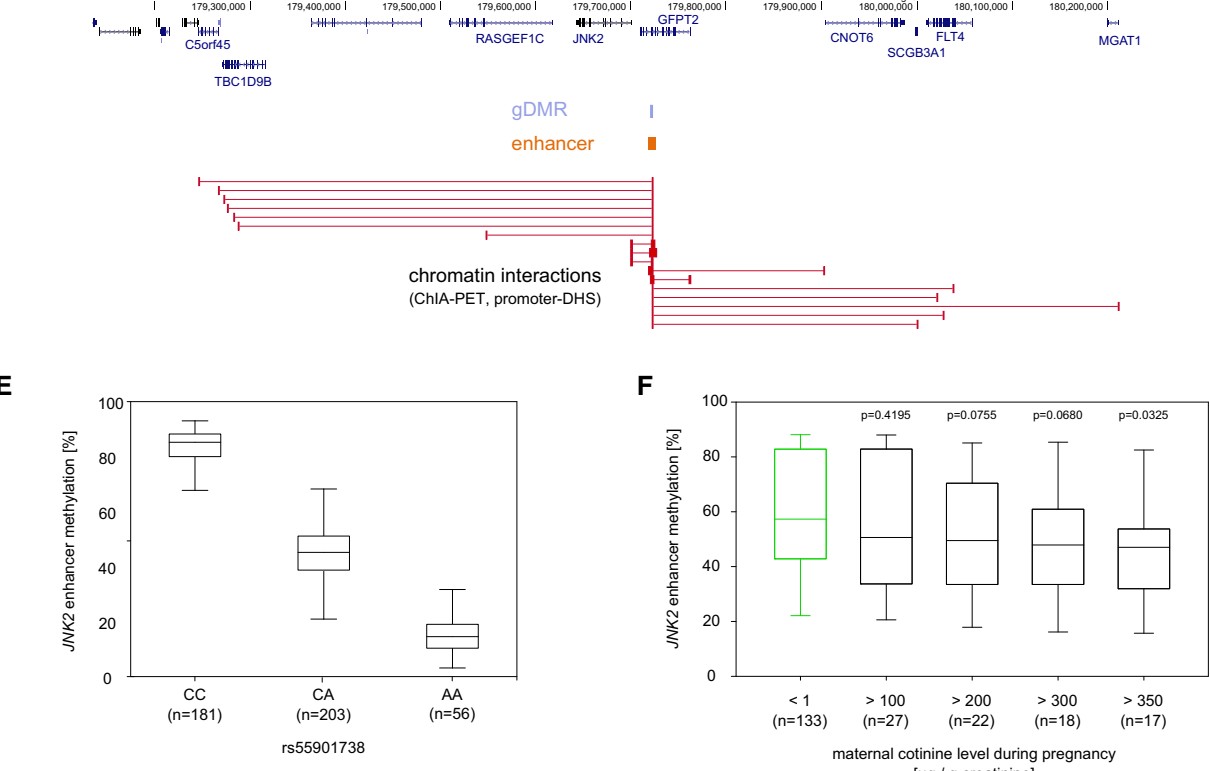

**Figure 6.**

◀

**Figure 6.   Validation of two representative examples of commuting enhancers.**

A   Extended locus of a commuting enhancer overlapping a ngDMR in the TMEM241 intron.

B   The hypermethylated DMR in the enhancer region identified in the cord blood of prenatally exposed children was confirmed by MassARRAY ($n$ = 471). The results show that concomitant with an increase in maternal cotinine levels, DNA methylation increases in this region.

C   Box plots compare methylation in the TMEM241 DMR evaluated by WGBS and by MassARRAY, respectively, for the same eight individuals. This comparison suggests that the relatively small effects size observed in the MassARRAY validation seems to be in part related to the weaker spread in methylation values obtained by this method.

D   Extended locus of the JNK2 enhancer, showing all predicted target genes by ChIA-PET and promoter-DHS interaction data.

E   A clear negative correlation between the genotype of rs55901738 (located inside the JNK2 DME) and the methylation level in this locus was observed in the discovery cohort ($N$ = 181/203/56).

F   The hypomethylated gDMR within the JNK2 enhancer region was similarly validated by MassARRAY in the entire cohort. Loss of DNA methylation in this region depends on maternal urine cotinine levels and decreases steadily with an increase in maternal urine cotinine concentrations. For children of heavy smoking mothers (cotinine level > 350 μg/g creatinine) we observe a methylation difference of over 10% compared to non-smoking mothers.

Data information: (B, C, E, F) Box plots visualize $25^{th}$ to $75^{th}$ percentile, line indicates the median, whiskers represent the non-outlier range.

findings that in the LISA cohort, the urine cotinine levels in children also reversely correlated with JNK2 enhancer methylation (Fig EV9C). To test whether DNA demethylation of the JNK2 enhancer is causatively related to tobacco smoke exposure, we applied an *in vitro* model based on peripheral blood mononuclear cells (PBMCs). Following a four-day exposure period with a cigarette smoke extract, we observed a decrease in enhancer methylation of 11.1% $\pm$ 7.6% (SD) in seven out of eight donors tested, an effect comparable to the reduction in DNA methylation in relation to urine cotinine levels (Fig EV9D).

**Link between "commuting enhancer" deregulation and phenotype development**

Finally, we aimed to elucidate whether differential DNA methylation in commuting enhancers was linked to a phenotype in the children. Since the function of TMEM241 is yet unknown, we focused this follow-up analysis on the JNK2 enhancer region. JNK2 is a member of the WNT signaling cascade, which has been implicated in a variety of tobacco smoke-induced lung diseases including COPD and lung cancer (Ying & Tao, 2009; Wang *et al*, 2011). Therefore, deregulation in the JNK2 enhancer might contribute to the development of adverse respiratory symptoms, although a link between lung disease development and JNK2 has not yet been described. Maternal smoking during pregnancy has been described as one of the contributing factors of impaired lung function and an increased risk of asthma and wheeze development in children (Burke *et al*, 2012). Wheezing, a whistling sound produced in the airways during breathing, is widely recognized as an early indicator of such an impaired lung function affecting a considerable number of children already in the first years of life. About 22–46% of children showing wheezing symptoms in early childhood are diagnosed with asthma later in life (Martinez *et al*, 1995).

We observed a hypomethylation in the JNK2 enhancer region related to tobacco smoke exposure in both mothers and newborn children (Table EV3). Additionally, histone modifications in this enhancer were also clearly linked to the smoking status. While in non-smoking mothers and their children a repressed or void chromatin state was present, smoking was associated with an active chromatin state in this region (Fig EV10A). In contrast, histone modifications in the promoter of JNK2 were indicative of an active state in all samples irrespective of their smoking status.

To further characterize the functional consequences of DNA hypomethylation in this enhancer region, we evaluated its

association with JNK2 transcription. DNA methylation and transcription correlated slightly but significantly in the entire cohort already at time of birth ($R$ = −0.11, $P$ = 0.044). Separating children from non-smoking ($R$ = −0.06, $P$ = 0.56) and smoking ($R$ = −0.46, $P$ = 0.034) mothers revealed that the overall significance was driven by the small subpopulation of children from smoking mothers (Fig EV10B) further supporting a combined effect of genotype and environment in transcriptional regulation of JNK2. In four-year-old children, the transcription of JNK2 is again slightly but significantly correlated with JNK2 enhancer methylation ($R$ = −0.16, $P$ = 0.04). Furthermore, blood levels of the pro-inflammatory cytokine IL-8, a downstream target of JNK2, correlated with JNK2 expression (Fig EV11A and B) (Holtmann *et al*, 1999; Hoffmann *et al*, 2002). Moreover, JNK2 transcription as well as IL-8 protein expression was significantly elevated in children developing wheezing symptoms from year four on compared to those who never showed any wheezing or other respiratory symptoms (Fig EV11B).

Similar to what we observed when comparing children of smoking and non-smoking mothers, children developing wheezing symptoms after the age of four (late-onset wheeze) show a DNA hypomethylation of the JNK2 enhancer. The association between JNK2 enhancer DNA demethylation and late-onset wheeze was apparent already at birth ($\Delta_{\text{Methylation}}$ = −17%) and remained stable until the age of four ($\Delta_{\text{Methylation}}$ = −14%) (Fig 7A). To control for possible confounding factors influencing the risk of wheeze (gender, number of siblings, the presence of a cat in the household, parental history of atopy, smoking during pregnancy, and school education), we performed a logistic regression analysis. These results confirmed that DNA hypomethylation in the JNK2 enhancer region was significantly associated with a higher risk of wheeze (Fig 7A). The resulting odds ratio (OR) of 1.40 implicates a risk increase of 40% per 10% methylation loss. Thus, children with full loss of methylation in this region at time of birth would have an almost 400% increase in risk to develop wheezing symptoms later in life. In the independent validation cohort (LISA), we confirmed that loss of DNA methylation in the JNK2 enhancer is associated with an increased risk for the development of wheezing symptoms in four-year-old children with an adjusted OR of 1.48 (CI:1.07–2.05; Fig 7A).

To further assess the role of JNK2 in the development of lung disease, we investigated the effect of JNK2 deficiency in a mouse asthma model. Sensitization and challenge of JNK2$^{-/-}$ mice with ovalbumin (OVA) resulted in markedly reduced airway inflammation shown by substantially less inflammatory infiltrates in the peribronchial and perivascular regions and a decreased mucous

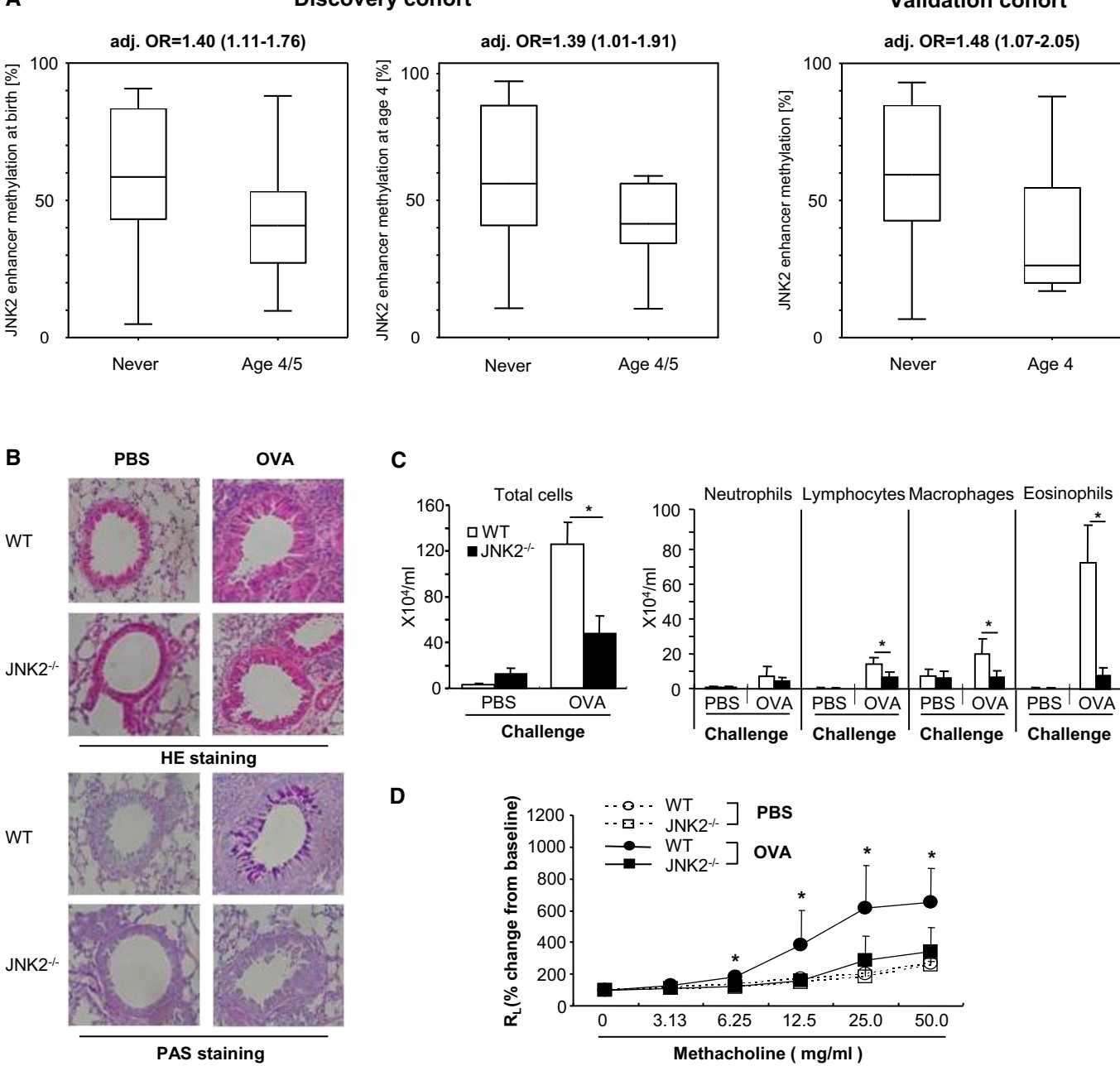

**Figure 7.  JNK2 is linked to lung disease.**

A    In both the discovery and the validation cohort, we show that loss of DNA methylation in the JNK2 enhancer is associated with an increased risk for the development of wheezing symptoms in four-year-old children. Odds ratios are calculated using logistic regression models adjusted for gender, number of siblings, the presence of a cat in the household, parental history of atopy, school education, and smoking during pregnancy (adj. OR). In 4-year-old children, regression models were additionally adjusted for postnatal smoking (infantile urine cotinine concentrations, box plots vizualize 25/75 percentile and median, whiskers represent non-outlier range).

B–D  JNK2 knockdown leads to a diminished asthma response in OVA-sensitized mice. (B) HE- and PAS-stained lung sections of OVA-sensitized JNK2$^{-/-}$ mice show reduced cell infiltration and mucous gland hyperplasia compared to wild-type controls. (C) The number of eosinophils, lymphocytes, and macrophages in BAL fluid is significantly lower in JNK2$^{-/-}$ mice, (D) which is accompanied by a decrease in AHR compared to OVA-sensitized WT mice (*$P < 0.05$ by Student's *t*-test, data are shown as mean $\pm$ SD ($n = 10$–15) of three independent experiments).

gland hyperplasia in the airway epithelium compared to OVA-sensitized wild-type (WT) animals (Fig 7B). Accordingly, the accumulation of eosinophils, lymphocytes, and macrophages in the

bronchoalveolar lavage (BAL) fluid of JNK2$^{-/-}$ mice was clearly diminished (Fig 7C). Furthermore, the methacholine-induced airway hyperreactivity (AHR), measured as lung resistance (RL), was

significantly reduced in JNK2$^{-/-}$ mice compared to WT controls (Fig 7D). Therefore, JNK2 deficiency leads to a diminished asthma phenotype, which supports the findings in our cohort confirming that JNK2 is directly involved in the development of lung disease. Taken together, we link loss of DNA methylation in the JNK2 commuting enhancer with the occurrence of impaired lung function mediated through differential expression of its target gene JNK2, a thus far unknown key player in the development of lung disease.

## Discussion

How prenatal exposure to smoking and other factors affect the newborn's epigenome and to which extent the resulting changes are stably maintained has been a central question for epigenetic medical research. Here, we acquired comprehensive genome-wide maps of DNA methylation changes linked to smoking and combined them with histone modification profiles and gene expression analysis. This represents an entirely different approach compared to previous EWAS focusing exclusively on methylation changes of a smaller subset of all CpGs genome wide. We demonstrated for the first time within a longitudinal human cohort study that *in utero* exposure to tobacco smoke affects the global epigenome of both mothers and children in regulatory elements and that this epigenetic modulation is stably maintained. Interestingly, while the unborn child is exposed to similar environmental challenges as its mother, the response on the DNA methylation and histone modification level is quite distinct. Apart from potential age-related epigenetic pattern, this could be due to the specificity of the placental barrier, which potentially leads to higher exposure of the unborn child to components of tobacco smoke as nicotine than the mother (Luck *et al*, 1985).

DNA methylation and enhancer activity are linked as demonstrated in a number of studies (Aran & Hellman, 2013),(Hon *et al*, 2013; Mukamel & Tanay, 2013). Notably, a preference of tissue specific DMRs at distal *cis*-regulatory elements was reported with some of these enhancers being dormant in adult tissues but active during embryonic development (Hon *et al*, 2013). The mechanistic details of the cross talk between DNA methylation, histone modifications, and enhancer RNA transcription are currently unclear but might involve links between DNA and histone modifications (Bergman & Cedar, 2013) as well as modulation of transcription factor binding due to DNA (hydroxy)methylation-directed nucleosome positioning (Teif *et al*, 2014). It should be noted that our results on the functional relevance of deregulated epigenetic signaling at intragenic enhancers and its link to disease risk could only be obtained from individualized epigenome annotations in addition to cell line-derived maps like histone modification ChIP-seq and DNase I hypersensitivity data from consortia like ENCODE or Roadmap.

Prenatal tobacco smoke exposure has been identified as one of the important risk factors for impaired lung function later in children's life. Accordingly, we specifically focused on potential links between DNA methylation changes and the development of respiratory diseases in children. Supported by our *in vitro* results, our epidemiological data indicate a combined impact of tobacco smoke and the genotype on DNA methylation of the JNK2 commuting enhancer located in the GFPT2 gene and show that DNA methylation loss and gain of activating histone marks within this commuting enhancer correlates with an increase of JNK2 gene expression. Loss of DNA methylation in the JNK2 enhancer was furthermore associated with an increased risk for children to develop wheezing symptoms later in their life. A mouse asthma model corroborated the functional relevance of our epigenome analysis. The JNK2$^{-/-}$ mice exhibited markedly reduced airway inflammation and airway hyperreactivity compared to WT controls suggesting a direct role for JNK2 in the development of lung disease. Further evidence for the regulatory role of JNK2 comes from a recent study demonstrating enhanced airway inflammation and impaired lung function due to a JNK2-triggered loss of the regulatory function of naturally occurring regulatory T cells (Joetham *et al*, 2014). Regulatory T cells were identified as important regulators of allergic lung diseases as they are recruited from the blood to allergen-challenged lungs inhibiting asthmatic responses. Since cross talk between activated immune cells from the blood and lung tissue was shown to contribute to the development and exacerbation of the local inflammatory response in the lung tissue (O'Donnell *et al*, 2006), changes of JNK2 activity in blood cells are likely to reflect corresponding changes in cells infiltrating inflamed lung tissue.

For the first time, we distinguished between genotype- and environment-related effects on the epigenome in a genome-wide approach. We showed that in both mothers and children, a significant proportion of the DMRs appear to be driven by nearby genotype effects (7005/8409 in children, 7790/9743 in mothers). However, this also means that around 20% of the DMRs are not affected by nearby SNPs, and might be the result of environmental exposure to tobacco smoke. Interestingly, the enrichment of this set of ngDMRs in enhancers is considerably more significant than the enrichment seen by considering all DMRs, indicating that ngDMRs indeed represent a separate class of elements. We also showed that differential methylation in annotated chromatin regions has an effect on gene expression, depending on the type of chromatin marks present in the locus. In particular, differential methylation in enhancer regions correlates negatively with expression, more than in promoter regions or in void regions. Hence, modulation of the enhancer activity through variable DNA methylation conditioned by non-genotype effects leads to a differential regulation of downstream genes (Aran *et al*, 2013; Blattler *et al*, 2014). The fact that enhancers overlapping with DMRs have more target genes compared to the full set of enhancers indicates that differential methylation is targeted at regulatory hubs. While histone modifications might also be under the control of the genotype (Kasowski *et al*, 2013; Kilpinen *et al*, 2013; McVicker *et al*, 2013), our focus was specifically on alterations in the DNA methylation pattern.

Although we focused our validation experiments on enhancer regions overlapping with DMRs, we do not claim that only epigenetic deregulation in enhancer regions are of biological relevance. We decided to focus on enhancers to emphasize the fact that regulatory regions are more frequently epigenetically deregulated by tobacco smoke exposure than other genomic regions. However, DMRs in other genomic regions could potentially also be related to a particular phenotype by enhancer-independent mechanism. A prominent example is the hypomethylated cg19859270 in the GPR15 gene body found in active and former smokers with a delta methylation of only 1–2%. As we showed in a previous study (Bauer *et al*, 2015), this very minor methylation change, identified in whole blood DNA samples, is related to the expansion of a small subset of GPR15 expressing T cells potentially involved in lung inflammation. This

example shows that even very minor methylation changes observed in whole blood samples might be of strong biological relevance.

Non-genetically associated differential DNA methylation was shown to be faithfully conserved over years in children's life. In the LINA cohort, only a relatively small number of children were exposed to tobacco smoke prenatally but not after birth. In many instances, however, children prenatally exposed to tobacco smoke remain exposed after birth. Accordingly, it was difficult to evaluate whether persistence of this environmental exposure after birth, albeit at a largely reduced effective concentration, could contribute to the high stability of induced DNA methylation changes in children over years after birth. Another important question is if other maternally or paternally transferred environmental challenges such as psychological stress, pollutants, and nutrition that are also monitored as part of the LINA study would have synergistic or independent effects on the children's epigenome. Thus, we anticipate that further exciting links between environmental factors and epigenetic deregulation will be revealed from the LINA dataset in continuation of the approach introduced here.

Apparently, the design of our study differs from conventional EWAS starting with a large sample set but focusing solely on DNA methylation changes in preselected CpG sites covering at best 5% of the genome. Although our WGBS approach was based initially only on 8 versus 8 mother–child pairs, we were able to confirm differential methylation at single CpG sites identified in those large EWAS (e.g., AHRR cg05575921, $\Delta_{Methylation} = -9.8\%$, *P*-value=0.007; MYO1G cg12803068, $\Delta_{Methylation} = 12.1\%$, *P*-value = 0.038; GFI1 cg18146737, $\Delta_{Methylation} = -16.9\%$, *P*-value = 0.05). Even though WGBS is not yet feasible for larger cohorts with hundreds or thousands of individuals, results obtained on a smaller sample set are representative as shown in the validation analysis (TMEM241 and JNK2 enhancer). The major advantage of this WGBS-based approach is the opportunity to study important genomic regions beyond promoters largely covered by DNA methylation arrays. Note that regulatory elements in particular enhancers are largely missing on these commonly used arrays. Thus, we were able to show for the first time that environmental exposure deregulates particularly regulatory regions of the genome. Moreover, by combining DNA methylation, histone modification, and gene expression analyses, we showed that epigenetic perturbation by environmental exposure is functionally translated and potentially linked to phenotype development. Thus, the depth and breadth of our analyses goes well beyond any earlier environmental epigenetics study.

## Materials and Methods

### Description of the cohorts

For this study, two population-based cohorts were employed. As discovery cohort, we used the prospective mother–child cohort, LINA (Lifestyle and environmental factors and their Influence on Newborns Allergy risk). For this cohort, 629 mother–child pairs (622 mothers and 629 children; seven twins) were recruited between May 2006 and December 2008 in Leipzig, Germany, to investigate the pre- and postnatal influences of lifestyle and environmental factors on the immune system of the newborn and the disease risk of the child later in life. Mothers suffering from

immune or infectious diseases during pregnancy were excluded from the study. Blood samples were obtained from mothers at the 36th week of gestation and cord blood at delivery (for details see Herberth *et al*, 2010, 2011). Longitudinal blood samples at year one till five were taken from both children and mothers (Fig EV1). During pregnancy, standardized questionnaires were recorded, as well as at each child's birthday (see Appendix Supplementary Methods). All questionnaires were self-administered by the parents. Maternal smoking was determined based on the questionnaire response and urine cotinine levels > 100 µg/g creatinine (see Appendix Supplementary Methods). During annual clinical visits, blood samples were obtained from children and mothers. Participation in the study was voluntary, and informed consent was obtained from all participants. The study was approved by the Ethics Committees of the University of Leipzig (046-2006, 160-2008, 160b/2008, 144-10-31052010, 113-11-18042011).

As validation cohort, we used a further prospective birth cohort, LISA (Life style–Immune System–Allergy). For LISA, a total of 2443 healthy neonates born in the German cities Munich and Leipzig were recruited between December 1997 and January 1999. The study design, blood sampling, and the questionnaire items used for the description of children's disease outcomes and confounding variables were comparable between LINA and LISA (Lehmann *et al*, 2002). The study was approved by the Ethics Committees of the University of Leipzig and Munich (206/2003). Written informed consent was obtained from the parents of all children.

### Whole-genome bisulfite sequencing

Whole blood samples were obtained from eight smoking and eight non-smoking mother–child pairs (see Appendix Supplementary Methods and Fig EV1). Libraries were prepared using the TruSeq DNA Sample Prep Kit v2-Set A (Illumina Inc., San Diego, CA, USA) according to the manufacturer's instructions. Adapter-ligated libraries were treated with bisulfite and PCR-amplified (see Appendix Supplementary Methods and Table EV10). Sequencing on HiSeq2000 (101-bp paired-end) was performed using standard Illumina protocols and the 200-cycle TruSeq SBS Kit v3 (Illumina Inc., San Diego, CA, USA).

### DMR calling and annotation

The bsseq v0.10 package (Hansen *et al*, 2012) for R statistical software v3.0.0 was used to smooth bisulfite sequencing data and call candidate DMRs. Because of our high average CpG coverage, we performed smoothing on a small window size with minimum 11 CpGs (ns = 11) and a minimum total width of 1 kb (*h* = 500), breaking the smoothing if gaps between CpGs exceeded 2 kb (maxGap = 2000). We calculated average raw methylation levels of each DMR and sample and performed a moderated *t*-test as in SAM statistics (significance analysis of microarrays, Tusher *et al*, 2001; R-package siggenes v1.36.0, Schwender, 2012) to assign *P*-values to each of the DMR. SAM *P*-values are low for DMRs that are consistently different between the groups. Based on the *P*-value (pSAM < 0.1) and the level of mean methylation changes ($\Delta_{Methylation} > 0.1$ in both raw and smoothed data), we filtered and ranked the DMR list for downstream analyses. We conducted false discovery rate analysis (FDR) based on permutation analysis

of DMRs before and after SAM filtering with thresholds for $\Delta_{\text{Methylation}}$ (1–25%). The obtained results suggest that a cutoff of 10% offers an optimal balance between high sensitivity and medium specificity, which is required when comparing groups of healthy individuals (see Appendix Supplementary Methods).

Initial genomic annotation of DMRs to the nearest TSS was obtained with the annotate Peaks script of the HOMER tools software package (Heinz *et al*, 2010) to genome version hg19. For calculating the significance of enrichment of DMRs in certain regions of interest (ROIs), we performed a random shuffling approach. First, regions of the same size as the DMRs were randomly sampled from the whole genome. In a second step, the number of overlaps of these random regions with the ROIs was calculated using BEDTools (Quinlan & Hall, 2010). This procedure was repeated 1000 times to determine the empirical null distribution. This empirical distribution was used to estimate the one-sided upper-tail *P*-values for enrichment. We calculated the fold change as the number of overlaps of the DMRs with the ROIs divided by the average number as determined by the randomizations.

### ChIP-seq assays and peak calling

ChIP-seq of histone modifications was performed in six mother–child pairs including three smoking and three non-smoking mothers and their children using standard ChIP-seq protocols (see Appendix Supplementary Methods). Regions of the genome exhibiting significant enrichment of histone modifications were identified using SICER v1.3 (Zang *et al*, 2009) and MACS v 1.4.1 (Zhang *et al*, 2008). Reads were aligned as outlined by Feng *et al* (2012). MACS was used to call peaks setting the histone modification alignment file as the treatment and the H3 SAM file as the control files. A threshold of $P \leq 1e\text{-}5$ was used to identify significant peaks. Additional peaks were called using SICER on all histone modifications with H3 as a control, removing all duplicate reads. The MACS and SICER peak calls were merged to maximize sensitivity. The peak calling summary is shown in Table EV1B.

### Genome segmentation and chromatin annotation

The chromatin was segmented and annotated using a multivariate Hidden Markov Model, ChromHMM (Ernst & Kellis, 2012). We trained a ChromHMM model over the four histone modification marks (H3K4me1, H3K27ac, H3K27me3, and H3K9me3) across all samples, which was subsequently used to segment the genome of each individual. The model was learned using the merged peak calls from MACS and SICER as the binarized input and allowing for a maximum of 400 iterations. We generated models with 5–16 states and decided to use 16 states as this captures all combinations of the four histone modifications so that rare chromatin states are also represented (Fig 4B). Each chromatin state was labeled based on a biologically interpretable name corresponding to the co-occurring histone marks. Chromatin states were merged over subsets of the data covering mothers, children, and their smoking and non-smoking subsets, where at least two samples had a genomic locus labeled to be of a particular chromatin state. To annotate promoter-associated marks, we identified all features overlapping with a RefSeq TSS, directly neighboring the overlapping feature with the TSS and its direct neighbors, and all features 400 bp from these,

and annotated them as "TSS associated". The remaining was labeled as "not TSS associated". Active regulatory elements were identified by merging three active states (states 1, 2, and 3, see Fig 4) into a common meta-state. Active regulatory elements, which were not TSS associated, were defined as enhancers. A repressed meta-state was defined by merging states 12, 13, and 14.

### RNA sequencing and analysis

RNA-seq libraries were built using the Illumina TruSeq RNA Sample Preparation Kit v2 (Illumina Inc., San Diego, CA, USA) according to the manufacturer's instructions. The final libraries were validated using Qubit (Invitrogen) and Agilent 2100 Bioanalyzer (Agilent Technologies). Sequencing on Illumina HiSeq2000/2500 (101-bp paired-end) was performed using standard protocols and the 200-cycle TruSeq SBS Kit v3 (see Appendix Supplementary Methods). RNA sequences were aligned to hg19 reference genome using the STAR alignment software (Dobin *et al*, 2013), with Gencode v19 as the transcriptome annotation. Counts of reads mapped to exons were estimated by htseq-count (Anders *et al*, 2015). For downstream analysis, raw counts were normalized by the Voom method in the limma package (Smyth, 2005) and the Combat method in the sva package (Jeffrey *et al*, 2014) was applied to correct for possible batch effects.

### Correlation analysis between DMRs and expression of target genes

We calculated the Spearman correlation between the mean DMR methylation and the target gene expression. For DMRs containing multiple CpG sites, mean methylation across CpG sites was calculated per sample. DMRs were first intersected with different chromatin states inferred from ChIP-seq data. For DMRs overlapping with void or TSS-associated states, the closest gene was used as the target gene. For DMRs overlapping with repressed or non-TSS-associated states, we used the predicted target gene from public interaction datasets (ChIA-PET and promoter–DHS interactions, see Appendix Supplementary Methods). For DMEs (DMRs overlapping with enhancers), we used the union of the target genes of the DMR and the overlapping enhancer. For each DMR-target gene pair, the significance of the correlation was calculated by the Spearman correlation test and the cutoff for significance was set to 0.05.

### Validation by MassARRAY methylation analysis and qPCR

Quantitative DNA methylation analyses of the enhancers in the TMEM241 and GFPT2 genes was performed using Sequenom's MassARRAY platform. Bisulfite-treated libraries were PCR-amplified. The PCR product was *in vitro* transcribed and cleaved by RNase A using the EpiTyper T Complete Reagent Set (Sequenom, Hamburg, Germany) and subjected to MALDI-TOF mass spectrometry analysis to determine methylation patterns as previously described (Ehrich *et al*, 2008). DNA methylation standards (0, 20, 40, 60, 80, and 100% methylated genomic DNA) were used to control for potential PCR bias.

For qPCR, total RNA was prepared from fresh cord blood by using peqGold RNA Pure (peqlab, Erlangen, Germany) and from blood collected in PAXgene Blood RNA Tube of year four by PAXgene

Blood RNA Kit (Qiagen, Hilden, Germany), according to manufacturer's instructions. The cDNA synthesis was carried out with 5 μg of RNA by using ImProm-II™ Reverse Transcription System (Promega, Mannheim, Germany). Gene expression was measured using the 96.96 Dynamic Array or FLEXsix Integrated Fluidic Circuits (IFCs) (Fluidigm, San Francisco, CA, USA). Intron-spanning primers were designed (see Appendix Supplementary Methods). Gene expression values were determined by using the $2^{-\Delta\Delta CT}$ method (Livak & Schmittgen, 2001) with *GAPD* and *GUSB*, as reference genes and normalized to the lowest measured value.

## Murine asthma model

JNK21$^{-/-}$ mice (C57BL/6 background) and control wild-type mice (WT) were bred and maintained at the animal facility of the Tokyo Medical University. Experiments were approved by the Ethical Committee of Animal Experiments of the Tokyo Medical University. JNK2$^{-/-}$ and C57BL/6J WT mice were sensitized and challenged with ovalbumin (OVA) and assayed for airway inflammation and airway hyperreactivity (AHR) as described before (Takada *et al*, 2013) and in the Appendix Supplementary Methods.

## Data availability

Next-generation sequencing data have been deposited at the European Genome-phenome Archive (EGA, http://www.ebi.ac.uk/ega/) hosted by the EBI, under accession numbers EGAS00001000455.

**Expanded View** for this article is available online.

## Acknowledgements

We would like to thank Stephan Wolf and Nicole Diessl at the DKFZ Genomics and Proteomics Core Facility for the excellent technical support and expertise. Further, we are grateful to Marion Bähr and Monika Helf who provided outstanding support in MassARRAY validation. We acknowledge very helpful discussions of cell lineage markers with Daniel Lipka and Soo-Zin Kim-Wanner and critical discussion of the manuscript with Christopher C. Oakes. We cordially thank the participants of the LINA study as well as Beate Fink, Anne Hain, Livia Sztraka, and Melanie Nowak for their excellent technical assistance and fieldwork. We are grateful to Martin von Bergen for providing urine cotinine concentrations. We also thank Pavel Komardin (Data management group at eilslabs) for the organization and implementation of all tasks involved in data submission to EBI. This work was supported by the Helmholtz cross program activity on Personalized Medicine (iMed) and the BMBF-funded eMED network PANC-STRAT (FKZ: 01ZX1305A). Major support came from the German Cancer Research Center—Heidelberg Center for Personalized Oncology (DKFZ-HIPO).

## Author contributions

MBa, ST, LT, DW, J M, OM MW, GH, MiS, CG, SWi, RE, and IL performed and/or coordinated experimental work. TB, NI, CH, LG, MBi, ZG KG, QW, SR, MBa, GH, ST, MaS, KR, and CL performed data analysis. MBo, SR, GH, UR-K, KJ, and IL collected data and provided proband materials. ET and JM performed the mouse experiments. TB, ST, NI, MBa, LG, QW, GH, LT, DW, CP, KJ, TP, SR, ZG, KR, C H, RE, and IL prepared the initial manuscript and figures. HS, CP, CH, RE, and IL provided project leadership. All authors contributed to the final manuscript.

## Conflict of interest

The authors declare that they have no conflicting interest.

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
