## [Review Process File · Molecular Systems Biology]

Environment-induced epigenetic reprogramming in genomic regulatory elements in smoking mothers and their children

Tobias Bauer, Saskia Trump, Naveed Ishaque, Loreen Thürmann, Lei Gu, Mario Bauer, Matthias Bieg, Zuguang Gu, Dieter Weichenhan, Jan-Philipp Mallm, Stefan Röder, Gunda Herberth, Eiko Takada, Oliver Mücke, Marcus Winter, Kristin M. Junge, Konrad Grützmann, Ulrike Rolle-Kampczyk, Qi Wang, Christian Lawrenz, Michael Borte, Tobias Polte, Matthias Schlesner, Michaela Schanne, Stefan Wiemann, Christina Geörg, Hendrik G. Stunnenberg, Christoph Plass, Karsten Rippe, Junichiro Mizuguchi, Carl Herrmann, Roland Eils and Irina Lehmann

Corresponding authors: Roland Eils, DKFZ; Irina Lehmann, Helmholtz Centre for Environmental Research - UFZ

Review timeline:

Submission date:	01 December 2015
Editorial Decision:	20 January 2016
Revision received:	02 February 2016
Editorial Decision:	08 February 2016
Revision received:	25 February 2016
Accepted:	04 March 2016

Editor: Thomas Lemberger

Transaction Report:

1st Editorial Decision

20 January 2016

Thank you again for submitting your work to Molecular Systems Biology. We have now heard back from the three referees who accepted to evaluate the study. As you will see, the referees find the topic of your study of potential interest. They raise however a series of concerns, which we would ask you to convincingly address in a revision of the present work.

The reviewers acknowledge the value of your extensive epigenetic dataset and the fact that the study presents a longitudinal analysis in pediatric samples. We feel therefore that we can consider this study for publication as a valuable resource for the community of researchers interested in epigenomics, chromatin and genome-epigenome-environment interactions. The recommendations provided by the reviewers are clear and refer to the following major points:

- the modest effect size observed should be explained and the used cutoff justified
- several aspects of the analysis should be clarified and the flow of the manuscript improved
- while we agree that it is interesting to analyze the JNK enhancer as an attempt to functionally assess the impact of differential methylation in 'commuting enhancers', the fact that this is a gDRM could be made clearer upfront. Reference to 'synergism' should be toned down (eg change to 'combined effect') in absence of data showing that combined genetic and environmental effects are larger than expected from their individual contributions.

REFEREE COMMENTS

Reviewer #1:

Summary

Bauer et al performed whole genome bisulfite sequencing, RNA-Seq, and ChIP-Seq of histone modifications in a large discovery cohort of mother/child pairs to test whether smoking in mothers is correlated with epigenetic, transcriptional, and functional affects in children from birth to 4 years of age. This is a fairly extensive and valuable dataset for the scientific community. They conclude that children of smoking mothers have: 1) widespread differences in DNA methylation that persist at least over several years of life and 2) an increased number of active enhancers. They then conclude that these epigenetic changes are causal to phenotypic changes in children of smoking mothers.

General remarks

The key feature of this manuscript is the extensive epigenetic dataset, which incorporates whole genome bisulfite sequencing rather than the more limited array platforms available. In addition, the addition longitudinal data, as well as histone modification and expression data distinguishes this manuscript from previous papers. Thus, this manuscript represents a key technical advance in the field.

However, the conceptual advances are limited. With this large amount of data, the authors logically attempt to tie DNA methylation changes with changes in histone modifications and expression changes. But the lack of global trends from integrative analysis weakens several of their key conclusions. Once published, I believe the manuscript would be of general interest to a wide audience of scientists studying epigenetics, environmental interactions, and gene regulation.

Major points

1. The sequencing coverage of bisulfite sequencing experiments is not well documented. The authors should include a table indicate the total number of sequenced, mapped, and monoclonal reads, as well as average coverage, for each WGBS library. The use of a large minimum size for DMRs (>1kb) would suggest that sequencing depth is quite low, which would limit their claim that their assay is "base pair resolution". To be clear, the authors should present simple analyses of the distributions of absolute percent CpG methylation genome-wide and in DMRs, as well as between-cohort differences in methylation at DMRs.
2. The authors make a clear link between smoking and changes in DNA methylation. It is also clear that the changes in methylation persist over time. They also present clear evidence that smoking is associated with an increase in active chromatin in children. However, the link between changes in methylation and changes in chromatin, at least as it is presented in this manuscript, is weak. For example, of the 66296 non TSS-associated regulatory elements, only 180 (<0.3%) overlap with ngDMRs. Since one can expect many overlaps just by random chance, the tiny fraction observed makes one wonder whether there is any true biological association in this dataset. Can the authors show, on some global level, that such overlaps are functionally significant? The authors attempt to show this in Figure EV6, but the analysis appears inconclusive.
3. On a similar note to Point 2, what happens to the vast majority of DMRs that do not overlap enhancers? Can they be related to the phenotype through an enhancer-independent mechanism? Or are they just noise? It's a shame that it appears they are just thrown away because they do not overlap with enhancers.
4. As the authors mention, the effect sizes in terms of methylation change are quite small, especially in the validation cohorts in Figure 6. Can the authors comment on what this means biologically? A given CpG in a diploid cell can either be 0%, 50%, or 100% methylated. What does observing a 10-15% change in methylation mean? Is this a technical issue that can be addressed with deeper sequencing or a real biological phenomenon?

5. The authors claim in the abstract that "By combining DNA methylation, histone modification, and gene expression analyses we show that epigenetic perturbation in enhancer regions is functionally translated." This language is far too strong for the evidence presented. To truly support this conclusion, the authors need to show convincing global evidence, which is not done. Rather, the authors use specific examples to support the conclusion, and the data from these examples are less than convincing. For example, the JNK2 enhancer is hypomethylated and gains active chromatin in the smoking cohort. If this observation were functional, one would expect a change in expression of JNK2 when comparing the smoking to non-smoking cohorts. Rather, it is shown that, only within the smoking cohort, there is a correlation between enhancer methylation and JNK2 expression. While true, this does not address the required point. The authors need to tone down their message or perform more convincing analyses.

6. Figure 7A shows that children developing wheezing have lower methylation of the JNK2 enhancer. It would be informative to incorporate smoking status into this panel. This would help to support the point that smoking is the driving variable behind wheezing and JNK2 enhancer methylation.

Minor points

1. The minimum methylation cutoff of 10% used for determining DMRs is arbitrary (and a bit low), and may be why the FDR is over 10%. For more stringent analysis, can the authors try another cutoff with an FDR closer to 1%?
2. It would be helpful to quantify the number of chromatin transitions observed in Figure 4C.
3. The authors' claim in the abstract that "Differential methylation preferentially targets a new class of intragenic 'commuting' enhancers" is misleading. How can the observation be "preferential" when so few DMRs overlap with "commuting" enhancers? The authors need to tone down their statements.
4. One underlying assumption of the manuscript is that smoking causes epigenetic changes which causes other changes including: histone modifications, expression, and phenotype. This chain of causality, while convenient, is not clearly supported. Again, the authors need to tone down their statements.

Reviewer #2:

In the manuscript "Environment-induced epigenetic reprogramming in genomic regulatory elements in smoking mothers and their children" by Bauer et al. an impressive amount of epigenomic data is generated in a cohort of smoking and non-smoking mothers and their children at 3 time points of early development: birth, 1 year and 4 years of age. WGBS and CHIP-seq for informative histone modifications are generated and presented. This is one of the largest epigenetic studies to date in terms of whole genome assessment of DNA methylation and histones for any environmental factor. Unfortunately, there is a lack of clarity and often of specifics in much of the analyses. There is no clear over of changes that occur specifically between mothers or between children for smokers and non-smokers, or during early childhood. Some of the desired analysis is presented for DMRs but these more specific comparisons are left out of later analyses that involve chromatin states. Related to this, most of the analysis is presented in a largely general overview even though the authors made the effort to define specific subcategories of DMRs and chromatin states. Again, clarity and specifics are needed throughout. The overall writing and paragraphical structure need improving.

The authors use a very modest change in methylation for DMR detection. 10% difference is required. Given they are measuring the whole genome, they should 1) use more than 5 promoters to determine % variation due to multiple cell types, and 2) determine how many DMRs exist within a group using the same parameters as between groups. A more stringent change of 20% may be more appropriate.

It is not explicit that DMRs called between smoking and non-smoking groups for children are time matched. Like much of the first few paragraphs setting up the experimental design, it is too vague. The authors should more clearly explain the comparisons made; how many DMRs came from each comparison; of DMR calls which are hyperDMRs and which are hypoDMRs.

It is unclear why the authors felt the need to smooth data give the high genome coverage.

In Tables EV2A and EV2B, why to the mean methylation levels at cell-specific promoters vary so much between mothers when comparing these two tables?

It is unclear if the authors were overly stringent by removing all DMRs that overlap or near a SNP (gDMRs). DMR calls are made by grouping neighboring CGs of similar differential status. It is, therefore, unclear that the SNP changed the local percent methylation. The approach as described in the text only seems applicable if all subjects in each group are of the same genotype and yet different between groups. Otherwise, if DMRs are removed because one individual in either group has a SNP, this seems very stringent. Since the group of children is small, how accurate is the correlations analysis described in the supplement. There are no charts, graphs, etc.. describing genotype to DMR relationship (meQTL).

The authors use ChIA-PET and other methods to find target genes. This should be better described in the main text. Moreover, this and the pathway analysis seem a little out of place in the first section of the paper which is largely a general description of the study design and global changes found (DMRs). It seems squeezed in, with limited dedicated text, to the end of this section. The authors should include this later when ChIA-PET is used again in the chromatin/DMR analysis, and when gene expression data is assessed.

These two sentences are unclear and illustrate a constant lack of clarity in just the first section of the results.

"About half of the pathways enriched by genes targeted by all DMRs overlapped between children and mothers (Table EV4A). In contrast, the vast majority of pathways enriched for target genes of ngDMRs were different for mothers and their children (Table EV4B) indicating a clear difference in environmental modulation of DNA methylation patterns in mothers and children."

By just saying "mothers and their children", this implies both smoking and non-smoking groups, and therefore they cannot conclude these are environmental differences. This could be developmental or age-associated differences - if one takes the sentence at face value. If the authors consider age an environmental influence contributing this, they should state it as such. In addition, these sentences are also returned as though it is a new paragraph. Two sentences do not make a paragraph. Frequently throughout there are returns but not blank lines separating sentences, which implies paragraphs. These are often made of up of one or two sentences. Please use a proper paragraph structure and/or indication of new paragraphs with a space or indentation. It is distracting.

5) Page 5: " we performed hierarchical clustering of all CpGs located within ngDMRs with sufficiently high coverage (n=4682 in children, n=7857 in mothers). Interestingly, the methylomes from the same individual at different years clustered perfectly (Figure EV3A and B) even after removing all genotype associated gDMRs prior to clustering. This suggests that a large fraction of specific DNA methylation sites is stably maintained over at least four years in children and over one year in mothers." What is sufficient coverage? After stating that ngDMRs were used for the analysis the authors state the clustering is grouped as expected "even after removing ... gDMRs". This relates to two points: 1) made early about how gDMRs are defined and if the genotypes are excluded; and 2) or why are gDMRs removed if the analysis is on ngDMRs?

6) Regarding Figure 3 and the associated text about changed in DMR methylation levels over time, the authors should provide a more quantitative figure that illustrates even the minor changes, such as box plots, violin plots and or average % methylation or change in % methylation averaged across all DMRs in each of the three categories of DMRs (all, ng and g), then again for qualitative, quantitative and unstable, in addition to Figure 3A and 3B.

Regarding the legend for Figure 3, how do know this is an enhancer region? The legend and piecharts for 3B are a bit hard to follow and made more concise. Maybe the percent cutoffs for the 3 color categories could be written directly on the figure in the legend box.

7) In the sentence " In summary, maternal smoking is associated with differential DNA methylation that is persistent in mothers and children up to four years after birth."

8) Regarding ChIP-seq and chromatin state analysis: Why did the authors determine 16 states and not use them, and only focus on meta-states? It might be misleading within the chromatin field to call states with active and repressive marks "bivalent" given the long history of that term as referring more specifically to H3K4me3+H3K27me3, and some of these states have 3 modifications (though I acknowledge those fall into either "active" or repressive" categories).

In the remaining analysis where states are overlapped with DMRs, this needs to be greatly expanded, and DMR should be split in the analysis between hyper- and hypo-DMRs to better understand how chromatin states are associated with specific changes in DNA methylation.

This reviewer is not a fan of the term "commuting" enhancers. Most enhancers loop in from afar. Another term would like best relay their point that they are in one gene but targeting another.

9) The authors need to dedicate a main figure to the distribution of DMRs and DMEs (and their various subcategories) at enhancer categories. Figure 5B does not include enhancer specifics.

10) Figure EV5 should be incorporated into a main figure. Additionally, a better description of differential expression and the relationship to DMRs, DMEs or chromatin changes at promoters and enhancers is needed. This will give a better perspective on how epigenetic changes caused by smoking are contributing to changes in gene expression. While some of this information is contained within Figure EV6, it needs to be elaborated on. For examples the authors have a made a point of creating so many types of DMRs (g-, ng- or DMEs; hyper and hypo are virtually absent from any descriptions/discussions), and chromatin states, but these often referred to in general terms in this section.

11) On page 9, the authors make several observations that need additional support or clarity. " Oct4 is expressed in both children and mothers in our cohort according to our RNA sequence data. Since the presence of an Oct4 binding site was related to hypomethylation (Zimmerman et al, 2013) the presence of such a binding site may contribute to the decrease of DNA methylation in individuals with the C->A genotype. In both the LINA discovery cohort (Figure 6E) and the LISA validation cohort (Figure EV7B), DNA methylation of the JNK2 commuting enhancer correlates with the genotype. Strikingly, the majority of children of smoking mothers in the LINA discovery cohort and of tobacco smoke exposed children in the LISA validation cohort display a C/A or A/A genotype in this position giving rise to an overall reduced methylation of the JNK2 commuting enhancer in smoking individuals (significant for maternal cotinine level > 350ug/g creatinine and for children's urine cotinine levels >40 ug/g creatinine; Table EV8A/B)." Is OCT4 relevant? Is it expressed in blood or tissues of interest? If the lack of methylation is due to loss of C at a CG, then is the creation of a binding site contributing the decrease; i.e. is the loss of a CG or the creation of a TFBS more important? If the children of smokers of the C/A or A/A genotype and demethylated, then this is genetics not environment and likely coincidental with smoking. Maybe the authors should limit their analysis to ngDMRs or provide statistically significant example where a genetic change at CGs leads to a point specific change in methylation, but these sites become increasing demethylated with smoking.

1st Revision - authors' response

02 February 2016

Reviewer #1

Summary

Bauer et al performed whole genome bisulfite sequencing, RNA-Seq, and ChIP-Seq of histone modifications in a large discovery cohort of mother/child pairs to test whether smoking in mothers is correlated with epigenetic, transcriptional, and functional affects in children from birth to 4 years of age. This is a fairly extensive and valuable dataset for the scientific community. They conclude that children of smoking mothers have: 1) widespread differences in DNA methylation that persist at least over several years of life and 2) an increased number of active enhancers. They then conclude that these epigenetic changes are causal to phenotypic changes in children of smoking mothers.

General remarks

The key feature of this manuscript is the extensive epigenetic dataset, which incorporates whole genome bisulfite sequencing rather than the more limited array platforms available. In addition, the addition longitudinal data, as well as histone modification and expression data distinguishes this manuscript from previous papers. Thus, this manuscript represents a key technical advance in the field.

However, the conceptual advances are limited. With this large amount of data, the authors logically attempt to tie DNA methylation changes with changes in histone modifications and expression changes. But the lack of global trends from integrative analysis weakens several of their key conclusions. Once published, I believe the manuscript would be of general interest to a wide audience of scientists studying epigenetics, environmental interactions, and gene regulation.

Major points

1. The sequencing coverage of bisulfite sequencing experiments is not well documented. The authors should include a table indicate the total number of sequenced, mapped, and monoclonal reads, as well as average coverage, for each WGBS library. The use of a large minimum size for DMRs (>1kb) would suggest that sequencing depth is quote low, which would limit their claim that their assay is "base pair resolution". To be clear, the authors should present simple analyses of the distributions of absolute percent CpG methylation genome-wide and in DMRs, as well as between-cohort differences in methylation at DMRs.

Most of the requested information is already been shown in Table EV1a. We apologize for the misunderstanding in interpretation of our data. First, the median size of a DMR is 222 and 223 bp for children and mothers respectively. Second, as one can see in Table EV1a sequencing coverage at CpG positions is reasonably well ranging between 18 and 46 fold. In the revised version, we added delta_methylation distributions for CpGs in all DMRs, ngDMRs and gDMRs complemented by information about the distribution with respect to their status of stability over time (quant./qual./unstable) for both children and mothers (see new Figure EV3B/D/E/F). Density plots of CpG methylation do not add much information to the other data shown so that we decided to not add these plots in the revised version. They are, however, included below for review purposes below.

2. The authors make a clear link between smoking and changes in DNA methylation. It is also clear that the changes in methylation persist over time. They also present clear evidence that smoking is associated with an increase in active chromatin in children. However, the link between changes in methylation and changes in chromatin, at least as it is presented in this manuscript, is weak. For example, of the 66296 non TSS-associated regulatory elements, only 180 (<0.3%) overlap with ngDMRs. Since one can expect many overlaps just by random chance, the tiny fraction observed makes one wonder whether there is any true biological association in this dataset. Can the authors show, on some global level, that such overlaps are functionally significant? The authors attempt to show this in Figure EV6, but the analysis appears inconclusive.

One of our main claims in the paper is that the DMRs and in particular the ngDMRs are strongly enriched in genomic elements with particular combinations of histone marks. However, we acknowledge that this was not presented clearly enough in the previous version. Therefore, we have replaced Figure 5B with the information previously contained in Table EV7, which shows this strong enrichment of DMRs in chromatin states such as active promoters but mostly active enhancers. Hence, while it is true that only a tiny proportion of all enhancers are targeted by differential methylation (0.03% as mentioned by the reviewer), a significant proportion of the DMRs overlap with this state (13%/11.1% of ngDMRs in children/mothers), resulting in a very significant overlap. We believe that our representation of this enrichment as barplot in the new Figure 5A should make this statement more clear.

As for the functional implications, indeed Figure EV6 makes the link between differential methylation in enhancers and changes in gene expression. We used DMRs overlapping the void state, which have no specific chromatin signature, as baseline. Those show little correlation with changes in expression. On the other hand, DMRs, which overlap either active or repressed chromatin states, show a clear increased association with expression changes, either positive for repressed elements or negative for active elements (Figure EV8B,C). We changed the text (paragraph "Transcriptional response of epigenetic reprogramming of regulatory genomic elements" on page 8 describing these results to clarify this aspect.

3. On a similar note to Point 2, what happens to the vast majority of DMRs that do not overlap enhancers? Can they be related to the phenotype through an enhancer-independent mechanism? Or are they just noise? It's a shame that it appears they are just thrown away because they do not overlap with enhancers.

It was far beyond the scope of this study to validate the potential relevance of every epigenetically deregulated region identified in smoking mothers and their newborn children. For the first time we show on a genome-wide scale that enhancer regions, and in particular commuting enhancers, potentially regulating a set of target genes, are affected in a stronger manner by tobacco smoke exposure than any other genomic region. To emphasize the importance of this result, we decided to select an enhancer region to demonstrate the potential link between epigenetic deregulation, transcriptional activation, and phenotype development. As expected, due to their interaction with promoter region and their impact on transcriptional regulation, we were able to demonstrate that enhancer deregulation on epigenetic level was related to the development of a clinical phenotype. For sure, it was not our intention to discard all other DMRs outside of enhancer regions. We fully agree that DMRs in other genomic regions might be also related to a phenotype by enhancer-independent mechanism. For example, we recently showed that hypomethylation of a CpG site

located in the gene body of the *GPR15* gene is related with an activation and expansion of a GPR15 expressing T cell subset reported to be involved in lung inflammation (Bauer et al, 2015). However, extending our functional analysis beyond the strongly enriched enhancer DMRs would have been well beyond the scope of the present study.

However, to address this point, we added the following part to the discussion: “Although we focused our validation experiments on enhancer regions overlapping with DMRs we do not claim that only epigenetic deregulation in enhancer regions might be of biological relevance. We decided to focus on enhancers to emphasize the fact that regulatory regions are more frequently epigenetically deregulated by tobacco smoke exposure than other genomic regions. However, DMRs in other genomic regions potentially could also be related to a particular phenotype by enhancer-independent mechanism. A prominent example is the hypomethylated cg19859270 in the *GPR15* gene body, found in active and former smokers with a delta methylation of only 1-2%. As we were able to show in a previous study (Bauer et al, 2015), this very minor methylation change, identified in whole blood DNA samples, goes back to the expansion of a small subset of GPR15 expressing T cells potentially involved in lung inflammation. This example impressively shows that even very minor methylation changes observed in whole blood samples might be of strong biological relevance. “

4. As the authors mention, the effect sizes in terms of methylation change are quite small, especially in the validation cohorts in Figure 6. Can the authors comment on what this means biologically? A given CpG in a diploid cell can either be 0%, 50%, or 100% methylated. What does observing a 10-15% change in methylation mean? Is this a technical issue that can be addressed with deeper sequencing or a real biological phenomenon?

We are grateful for the reviewer for this comment as we consider this an important point that should be better clarified in the revised version. In fact, there is a simple explanation for the observed methylation differences in a 10-15% range: the analysis of DNA-methylation in whole blood samples. The different blood cell populations show different methylation pattern and are potentially affected in a different manner by tobacco smoke exposure. Monocytes, for example, represent 5-10 % of the entire blood cells. Methylation changes less than 10% would be observed in whole blood, if only this cell population is affected. Several studies focusing on DNA methylation changes in the blood of smokers compared to non-smokers identified a highly significant, but only 1% methylation change in cg19859270 in the *GPR15* gene body. As mentioned above, we were able to show in a previous study (Bauer et al, 2015) that this very minor methylation change, identified in whole blood DNA samples, relates to the expansion of a small subset of GPR15 expressing T cells potentially involved in lung inflammation. This example impressively shows that even very minor methylation changes observed in whole blood samples might be of strong biological relevance. As mentioned in reply to comment no. 4 above, we added this more recent finding in the discussion of our revised paper.

5. The authors claim in the abstract that "By combining DNA methylation, histone modification, and gene expression analyses we show that epigenetic perturbation in enhancer regions is functionally translated." This language is far too strong for the evidence presented. To truly support this conclusion, the authors need to show convincing global evidence, which is not done. Rather, the authors use specific examples to support the conclusion, and the data from these examples are less than convincing. For example, the JNK2 enhancer is hypomethylated and gains active chromatin in the smoking cohort. If this observation were functional, one would expect a change in expression of JNK2 when comparing the smoking to non-smoking cohorts. Rather, it is shown that, only within the smoking cohort, there is a correlation between enhancer methylation and JNK2 expression. While true, this does not address the required point. The authors need to tone down their message or perform more convincing analyses.

We fully agree with the reviewer that our statement in the previous version of the paper was not sufficiently supported by data. We therefore significantly down-toned our statement in the revised paper and specifically related it to data shown in Fig. EV8: “Combined DNA-methylation, histone modification, and gene expression analyses indicate that differential methylation in enhancer regions is more often functionally translated than methylation changes in promoters.”

6. *Figure 7A shows that children developing wheezing have lower methylation of the JNK2 enhancer. It would be informative to incorporate smoking status into this panel. This would help to support the point that smoking is the driving variable behind wheezing and JNK2 enhancer methylation.*

We apologize for this misunderstanding. We do not claim that smoking is the driving variable behind wheezing and JNK2 enhancer activation. Our aim was to show that “differential DNA methylation in commuting enhancers was linked to a phenotype in the children”. We selected JNK2 overlapping with a ngDNR due to the known function of this gene suggesting a potential role in lung pathology. What we wanted to show is that in the case of JNK2 a) both the genetic background and smoking contribute to hypomethylation of the enhancer region and b) hypomethylation of the enhancer region is linked to the development of a lung phenotype in early childhood. Thus, either the genetic background or smoking or a combined effect of both might be drivers of the development of this phenotype. We have clarified this point in the text (see paragraph “Link between “commuting enhancer” deregulation and phenotype development” on page 10.

Minor points

1. *The minimum methylation cutoff of 10% used for determining DMRs is arbitrary (and a bit low), and may be why the FDR is over 10%. For more stringent analysis, can the authors try another cutoff with an FDR closer to 1%?*

This is an important point, which we specifically addressed in the revised version. Note that as discussed above (see answers to comment no. 3 above) we consider even very subtle methylation changes to be of high importance. Lowering the threshold beyond 10% would include such very subtle change, however, at the cost of dramatically increased FDR. Increasing the threshold beyond 10% would further decrease the FDR, however, at a very much reduced sensitivity. We conducted DMR calling at different thresholds (1%, 2%, 5%, 10%, 20%, and 25%) as suggested by this and the second reviewer to quantitatively assess an “optimal” cut-off for DMR calling that best balances between sensitivity and specificity by determining the false discovery rate (FDR) based on permutation analysis of DMRs before and after SAM-filtering. The results suggest that the cut-off of 10% provided the best balance between high sensitivity and medium specificity. We added a new Figure in the Expanded View (Fig. EV2) and an additional paragraph in the Methods section paragraph “DMR calling and annotation” on page 14.

2. *It would be helpful to quantify the number of chromatin transitions observed in Figure 4C.*

All the numbers of transitions and the proportion of the genome that show each kind of transition are given in Table EV6.

3. *The authors' claim in the abstract that "Differential methylation preferentially targets a new class of intragenic 'commuting' enhancers" is misleading. How can the observation be "preferential" when so few DMRs overlap with "commuting" enhancers? The authors need to tone down their statements.*

To address this point we show in the revised version (Fig. 5A) that overall enhancers are more often hit by differential methylation than any other analysed genomic region. Out of the enhancers, intragenic enhancers (of which 82% are commuting) are more prevalent than intergenic (about 2/3 vs. 1/3). The same statement holds true for the statistical significance level of those enrichments, which in mothers are most pronounced for intragenic enhancers. However, in line with the recommendation by this reviewer we changed this statement in the abstract: “Differential methylation is enriched in enhancer elements and targets in particular “commuting“ enhancers having multiple, regulatory interactions with distal genes. “

4. *One underlying assumption of the manuscript is that smoking causes epigenetic changes which causes other changes including: histone modifications, expression, and phenotype. This chain of causality, while convenient, is not clearly supported. Again, the authors need to tone down their statements.*

We agree with this reviewer that we have no evidence for this chain of causality. We therefore made several changes throughout the text to better reflect the quality of data presented in the manuscript (see in particular in subsection “Transcriptional response of epigenetic reprogramming of regulatory genomic elements” and in abstract).

Reviewer #2

1) In the manuscript "Environment-induced epigenetic reprogramming in genomic regulatory elements in smoking mothers and their children" by Bauer et al. an impressive amount of epigenomic data is generated in a cohort of smoking and non-smoking mothers and their children at 3 time points of early development: birth, 1 year and 4 years of age. WGBS and ChIP-seq for informative histone modifications are generated and presented. This is one of the largest epigenetic studies to date in terms of whole genome assessment of DNA methylation and histones for any environmental factor. Unfortunately, there is a lack of clarity and often of specifics in much of the analyses. There is no clear overview of changes that occur specifically between mothers or between children for smokers and non-smokers, or during early childhood. Some of the desired analysis is presented for DMRs but these more specific comparisons are left out of later analyses that involve chromatin states.

We agree with this reviewer that our study was designed to be very DMR centric. As such, our analysis of differential histone modification states is limited. Nevertheless, we present a detailed quantitative and statistical significance analysis of chromatin state transitions overlapping with DMRs, gDMRs and ngDMRs (Table EV6) and also show details on transitions of chromatin states between the two smoking/non-smoking groups (Figure 4C).

2) Related to this, most of the analysis is presented in a largely general overview even though the authors made the effort to define specific subcategories of DMRs and chromatin states. Again, clarity and specifics are needed throughout. The overall writing and paragraphical structure need improving.

We apologize for not having been very clear in the description of our work and the results in the earlier version of our manuscript. As described below, we undertook substantial effort to improve the description of our work throughout the text, changed the style of paragraphs, added additional data where required and modified some of our figures (e.g., Figure 5). We overall believe that thus the clarity of our paper has greatly been improved.

3) The authors use a very modest change in methylation for DMR detection. 10% difference is required. Given they are measuring the whole genome, they should 1) use more than 5 promoters to determine % variation due to multiple cell types, and 2) determine how many DMRs exist within a group using the same parameters as between groups. A more stringent change of 20% may be more appropriate.

We thank the reviewer for addressing this point. We followed this reviewer's advice and further improved the determination of blood cell populations based on published data for isolated cell populations (for detailed method description see Expanded View) and show now results for 7 blood cell subsets. Apart from this point we would not agree that focusing on more stringent methylation-change of 20% would be more appropriate. As mentioned above (see response to comment no. 3 of reviewer #1), even a very minor methylation change of 1-2 % measured in whole blood samples could be of strong biological relevance.

4) It is not explicit that DMRs called between smoking and non-smoking groups for children are time matched. Like much of the first few paragraphs setting up the experimental design, it is too vague. The authors should more clearly explain the comparisons made; how many DMRs came from each comparison; of DMR calls which are hyperDMRs and which are hypoDMRs.

We apologize again for not having been clear enough in the first version of our manuscript. We added additional explanations in the first and second subsection in the Results (see paragraph “Maternal smoking is associated with genome-wide DNA methylation changes that are different between mothers and their children” on page 4, and paragraph “DNA methylation changes due to maternal smoking are stably maintained over years of life” on page 5-6).

5) It is unclear why the authors felt the need to smooth data given the high genome coverage.

We applied a very local smoothing procedure, taking into account 11 CpGs as described in the Methods. The smoothing window is much smaller than the one used in the initial bsseq publication (smoothing window of 70 CpGs, (Hansen et al, 2011)), given our much higher coverage. However, smoothing allows reducing the effect of CpGs that might have a lower local coverage.

6) In Tables EV2A and EV2B, why do the mean methylation levels at cell-specific promoters vary so much between mothers when comparing these two tables?

We thank the reviewers for addressing this point. As mentioned above (see response to comment no. 3 of this reviewer), we have revised and improved our analysis resulting in more convincing and comparable results for mothers (see Table R1 below). For mothers, there is still a slight difference in blood cell composition between pregnancy and thereafter. This slight difference is caused by the altered immune regulation during pregnancy, which has been reported many times before. Nevertheless, we felt that showing results for both time points is misleading for the readers since DMR calling was only performed at the time of birth. Thus, in the revised manuscript only data for the first time point are shown. The table below is shown for review purposes.

Table R1:
Cell type distribution in maternal blood during pregnancy and one year after children's birth estimated by methylation signature

Cell type	marker	Smoking group ^a	Non-smoking group ^a	p-value ^b
Pregnancy				
Granulocytes	ACAD8	0.77 (0.10)	0.75 (0.04)	0.38
Monocytes	KIAA0930	0.12 (0.07)	0.10 (0.07)	0.83
T lymphocytes	CD3D, CD3G	0.23 (0.04)	0.19 (0.03)	0.38
CD8+ T cells	CD8A	0.06 (0.03)	0.05 (0.03)	0.34
CD4+ T cells	CD28	0.14 (0.05)	0.20 (0.06)	0.06
B lymphocytes	LILRB4	0.06 (0.04)	0.05 (0.04)	0.71
NK cells	KLRD1	0.12 (0.04)	0.11 (0.03)	0.49
Year one				
Granulocytes	ACAD8	0.71 (0.13)	0.59 (0.11)	0.40
Monocytes	KIAA0930	0.13 (0.06)	0.09 (0.04)	0.51
T lymphocytes	CD3D, CD3G	0.21 (0.05)	0.32 (0.06)	0.20
CD8+ T cells	CD8A	0.10 (0.09)	0.07 (0.01)	0.70
CD4+ T cells	CD28	0.25 (0.11)	0.32 (0.05)	0.70
B lymphocytes	LILRB4	0.10 (0.03)	0.07 (0.04)	1.00
NK cells	KLRD1	0.13 (0.04)	0.17 (0.07)	0.51

a. estimated based on the mean (standard deviation) methylation level of representative CpG sites in the promoter region (for CpG positions see Expanded view)

b. p-value calculated by the Mann-Whitney U-test

7) *It is unclear if the authors were overly stringent by removing all DMRs that overlap or near a SNP (gDMRs). DMR calls are made by grouping neighboring CGs of similar differential status. It is, therefore, unclear that the SNP changed the local percent methylation. The approach as described in the text only seems applicable if all subjects in each group are of the same genotype and yet different between groups. Otherwise, if DMRs are removed because one individual in either group has a SNP, this seems very stringent. Since the group of children is small, how accurate is the correlations analysis described in the supplement. There are no charts, graphs, etc.. describing genotype to DMR relationship (meQTL).*

We defined as gDMRs all DMRs for which there is a neighboring SNP (within +/- 5kb) which have a significant correlation (at 10% FDR) between the genotype and the methylation. Hence the presence of a neighboring SNP alone is not sufficient to qualify a DMR as gDMR, if there is no association between the genotype and the methylation. Regarding the significance, we applied a 10% FDR threshold on the correlation genotype/methylation; all DMRs showing an association with a smaller FDR are classified as gDMRs. Hence, this rather lenient threshold for gDMRs corresponds to a very stringent threshold for ngDMRs, as we rather have false negative ngDMRs than false-positives. We have clarified this in the Methods, and added a plot showing the relation between the correlation genotype/DMR and the delta-methylation for the gDMRs and ngDMRs (Fig. EV3A,C).

8) *The authors use ChIA-PET and other methods to find target genes. This should be better described in the main text. Moreover, this and the pathway analysis seem a little out of place in the first section of the paper which is largely a general description of the study design and global changes found (DMRs). It seems squeezed in, with limited dedicated text, to the end of this section. The authors should include this later when ChIA-PET is used again in the chromatin/DMR analysis, and when gene expression data is assessed.*

We have moved this part of the text as suggested to the section on transcriptional response.

9) *These two sentences are unclear and illustrate a constant lack of clarity in just the first section of the results.*

"About half of the pathways enriched by genes targeted by all DMRs overlapped between children and mothers (Table EV4A). In contrast, the vast majority of pathways enriched for target genes of

ngDMRs were different for mothers and their children (Table EV4B) indicating a clear difference in environmental modulation of DNA methylation patterns in mothers and children."

By just saying "mothers and their children", this implies both smoking and non-smoking groups, and therefore they cannot conclude these are environmental differences. This could be developmental or age-associated differences - if one takes the sentence at face value. If the authors consider age an environmental influence contributing this, they should state it as such. In addition, these sentences are also returned as though it is a new paragraph. Two sentences do not make a paragraph. Frequently throughout there are returns but not blank lines separating sentences, which implies paragraphs. These are often made of up of one or two sentences. Please use a proper paragraph structure and/or indication of new paragraphs with a space or indentation. It is distracting.

This sentence has been rephrased to better clarify that enrichment is always calculated with respect to genomic regions that are differentially enriched when comparing the smoking associated vs. non-smoking associated groups (paragraph "Transcriptional response of epigenetic reprogramming of regulatory genomic elements" on page 8. Please note that as such there is no enrichment for the smoking or non-smoking group, but only an enrichment, which refers to a comparison between the two smoking/non-smoking groups.

10) Page 5: "we performed hierarchical clustering of all CpGs located within ngDMRs with sufficiently high coverage (n=4682 in children, n=7857 in mothers). Interestingly, the methylomes from the same individual at different years clustered perfectly (Figure EV3A and B) even after removing all genotype associated gDMRs prior to clustering. This suggests that a large fraction of specific DNA methylation sites is stably maintained over at least four years in children and over one year in mothers." What is sufficient coverage? After stating that ngDMRs were used for the analysis the authors state the clustering is grouped as expected "even after removing ... gDMRs". This relates to two points: 1) made early about how gDMRs are defined and if the genotypes are excluded; and 2) or why are gDMRs removed if the analysis is on ngDMRs?

Sufficient coverage was defined as >10x. This is now clarified in the text (paragraph "DNA methylation changes due to maternal smoking are stably maintained over years of life" on page 5 and corresponding legend to Figure EV4). Further, we rephrased that sentence to avoid confusion between ngDMRs and gDMRs.

6) Regarding Figure 3 and the associated text about changed in DMR methylation levels over time, the authors should provide a more quantitative figure that illustrates even the minor changes, such as box plots, violin plots and or average % methylation or change in % methylation averaged across all DMRs in each of the three categories of DMRs (all, ng and g), then again for qualitative, quantitative and unstable, in addition to Figure 3A and 3B.

This aspect has also been brought up by reviewer #1. We now added violin plots as requested in Fig. EV3. Further details and plots are found in response to comment 1 by reviewer #1 above.

Regarding the legend for Figure 3, how do know this is an enhancer region? The legend and piecharts for 3B are a bit hard to follow and made more concise. Maybe the percent cutoffs for the 3 color categories could be written directly on the figure in the legend box.

We thank this reviewer for spotting this. We mistakenly described this region as enhancer. This mistake is now corrected in the revised version and the legend updated.

7) In the sentence "In summary, maternal smoking is associated with differential DNA methylation that is persistent in mothers and children up to four years after birth."

The question appears to be missing in this comment.

8) Regarding ChIP-seq and chromatin state analysis: Why did the authors determine 16 states and not use them, and only focus on meta-states? It might be misleading within the chromatin field to call states with active and repressive marks "bivalent" given the long history of that term as

referring more specifically to H3K4me3+H3K27me3, and some of these states have 3 modifications (though I acknowledge those fall into either "active" or repressive" categories).

We chose to use 16 chromatin states in our model so that every possible combination (2^4) of histone marks is covered. We only focus on the meta-states as the limited number of histone marks reduces the power of chromatin segmentation and can separate functional genomic elements due to missing histone marks. We use the meta-states to combine neighboring features into functional genomic units to overcome the problem with having too few histone marks. To illustrate this with an example, an active promoter would typically have H3K4me1/2/3 and H3K27ac and H3K9ac enriched (Reddington et al, 2013), but the co-localization of these marks is not at every histone, so by using meta-states one can combine the neighboring signals at such a promoter into one functional unit.

9) In the remaining analysis where states are overlapped with DMRs, this needs to be greatly expanded, and DMR should be split in the analysis between hyper- and hypo-DMRs to better understand how chromatin states are associated with specific changes in DNA methylation.

We agree with the reviewer that it is important to investigate the relationship between methylation and histone modifications. This cannot be done at the chromatin state level as this binarizes the chromatin signal so differential histone modification are lost. We have previously attempted to incorporate differential-histone-DMR analysis into the manuscript, but the results did not achieve a desirable level of global statistical significance. The main reason for the inability to achieve statistical significance is due to: the high noise in ChIP-seq data in general, the subtle differences between the smoking/non-smoking samples and the low number of samples for comparison (3 vs 3). So for our study we focus on using the histone marks to annotate the chromatin states rather than for differential analysis. However, we will report our findings to satisfy the curiosity of the reviewer and to show that the findings are both interesting and intuitive.

We have taken a DMR-centric approach to investigating this relationship. We find that any trends are only observable for DMRs with p value less than 0.01. When looking at the normalized read counts (calculated by the R package diffbind, normalizing to the H3 control and to the total library size) around hyper- and hypo-DMRs in smokers and non-smokers (extending by 500 bp to capture the signal of the neighboring nucleosomes). We see trends that H3K27ac and H3K27me3 are negatively correlated with methylation (please refer to Figure R1 below, which have been added for review purposes). The negative associations of H3K27ac is intuitive as these are active marks, but the negative association of the repressive H3K27me3 mark with DNA methylation has also been reported in literature (Reddington et al, 2013). It is interesting that we only observe these trends in the ngDMRs and not the gDMRS nor the combined DMR sets (data not shown). We did not see any clear pattern for H3K4me1 (data not shown).

We further investigated this looking into the odds-ratios of the differential histone marks in the DMRs. We selected only DMRs that have $p < 0.01$, and compare them to the direction of differential histone modification (please refer to Table R2). We observed a negative association of H3K27ac and H3K4me1 with DNA methylation in ngDMRs in both mothers and children. The negative association of H3K27me3 with DNA methylation was less clear. We also saw a positive association of H3K9me3 with DNA methylation for both gDMRs and ngDMRs.

We hope that this is sufficient indication for the reviewer that there are biologically meaningful associations between differential histone modifications and differential DNA methylation, but we believe it is out of the scope of this study as it requires a large cohort of samples and more histone marks to be sampled.

Figure R1. H3K27ac and H3K27me3 read occupancy in ngDMRs of children

Table R2:

Odds-ratios of the differential histone marks in the DMRs

Children	H3K4me1		H3K27ac		H3K27me3		H3K9me3	
	gDMRs	ngDMRs	gDMRs	ngDMRs	gDMRs	ngDMRs	gDMRs	ngDMRs
p<0.01 DMRs								
Odd-ratios	2,81	0,67	1,17	0,15	1,36	0,67	1,7	1,33

Mothers	H3K4me1		H3K27ac		H3K27me3		H3K9me3	
	gDMRs	ngDMRs	gDMRs	ngDMRs	gDMRs	ngDMRs	gDMRs	ngDMRs
p<0.01 DMRs								
Odd-ratios	1,26	0,60	0,94	0,69	0,67	1,19	1,51	1,27

This reviewer is not a fan of the term "commuting" enhancers. Most enhancers loop in from afar. Another term would like best relay their point that they are in one gene but targeting another.

We appreciate the suggestion for renaming commuter enhancer as relay enhancer. With the term "commuter" we wanted to stress the point that these enhancers commute between different genes rather than the effect of remote looping. As much as we like the alternative term relay enhancer we still believe that the term commuter enhancer is slightly more intuitive.

10) The authors need to dedicate a main figure to the distribution of DMRs and DMEs (and their various subcategories) at enhancer categories. Figure 5B does not include enhancer specifics.

We thank the reviewer for this comment, as it points to a confusing element in the previous version of the manuscript. As our analysis is mostly DMR-centered, we have replaced Figure 5B with a figure showing the enrichment of the various DMR categories in the different chromatin states defined from the ChIP-seq data, i.e. a DMR-centric analysis rather than an enhancer centric analysis, as was previously the case. This figure now shows the statistical enrichment and the percentage of DMRs in each of the chromatin categories in a much more intuitively understandable way. A comprehensive table with all the numbers is included as supplementary Table in the Expanded View

(Table EV7, as previously). We have moved the old part of Figure 5B to a supplementary table (Table EV8)

11) Figure EV5 should be incorporated into a main figure. Additionally, a better description of differential expression and the relationship to DMRs, DMEs or chromatin changes at promoters and enhancers is needed. This will give a better perspective on how epigenetic changes caused by smoking are contributing to changes in gene expression. While some of this information is contained within Figure EV6, it needs to be elaborated on. For examples the authors have made a point of creating so many types of DMRs (g-, ng- or DMEs; hyper and hypo are virtually absent from any descriptions/discussions), and chromatin states, but these often referred to in general terms in this section.

We thank the reviewer for underlining the importance of functional analysis of gene expression in the context of pathway enrichment analysis. In MSB style Expanded View figures are embedded into the online version of the paper the same way as main figures. As such, we decided to keep this figure as Expanded View figure. Following the advise of Reviewer 1, we moved the section on pathway enrichment of target genes of DMRs to the section on transcriptional response. Hence, we added to our general analysis on the link between differential methylation and changes in expression an analysis focused on the functional consequences of differential methylation in active regulatory elements (enhancers and promoters) in terms of coordinated changes in expression of functionally relevant pathways. In the same section, we also clarified which subset of DMRs we considered in the analysis, according to the recommendations of this reviewer. We have also made more explicit the fact that in the analysis of the link between differential methylation and expression we take the DMRs intersecting the void state as a baseline for correlation, and show that DMRs intersecting promoters, enhancers or repressed states show much more significant correlations compared to this ground state.

12) On page 9, the authors make several observations that need additional support or clarity. " Oct4 is expressed in both children and mothers in our cohort according to our RNA sequence data. Since the presence of an Oct4 binding site was related to hypomethylation (Zimmerman et al, 2013) the presence of such a binding site may contribute to the decrease of DNA methylation in individuals with the C->A genotype. In both the LINA discovery cohort (Figure 6E) and the LISA validation cohort (Figure EV7B), DNA methylation of the JNK2 commuting enhancer correlates with the genotype. Strikingly, the majority of children of smoking mothers in the LINA discovery cohort and of tobacco smoke exposed children in the LISA validation cohort display a C/A or A/A genotype in this position giving rise to an overall reduced methylation of the JNK2 commuting enhancer in smoking individuals (significant for maternal cotinine level > 350ug/g creatinine and for children's urine cotinine levels >40 ug/g creatinine; Table EV8A/B)."

Is OCT4 relevant? Is it expressed in blood or tissues of interest? If the lack of methylation is due to loss of C at a CG, then is the creation of a binding site contributing the decrease; i.e. is the loss of a CG or the creation of a TFBS more important? If the children of smokers of the C/A or A/A genotype and demethylated, then this is genetics not environment and likely coincidental with smoking.

Maybe the authors should limit their analysis to ngDMRs or provide statistically significant example where a genetic change at CGs leads to a point specific change in methylation, but these sites become increasing demethylated with smoking.

We again thank the reviewer for pointing out this lack of clarity. 1) Oct4 is expressed in both children and mothers, this is explicitly stated on page 9, in the second paragraph, 2) It is not a CpG destroying SNP but rather creates a putative binding site. This is stated in in the same paragraph on page 9. 3) We pointed out very clearly in the manuscript that this DMR has a strong genetic effect and we show this effect e.g. in Figure 6. We, however, also show data that there is an additive effect from the environment (see in vitro data in Figure EV 9D). We attempted to better clarify these points in the revised manuscript in the corresponding section "Validation of tobacco smoke exposure related differential methylation in the entire cohort" (2nd paragraph of this section on page 9).

References

Bauer M, Linsel G, Fink B, Offenberger K, Hahn AM, Sack U, Knaack H, Eszlinger M, Herberth G (2015) A varying T cell subtype explains apparent tobacco smoking induced single CpG hypomethylation in whole blood. *Clinical Epigenetics* 7: 1-11

Hansen KD, Timp W, Bravo HC, Sabunciyan S, Langmead B, McDonald OG, Wen B, Wu H, Liu Y, Diep D, Briem E, Zhang K, Irizarry RA, Feinberg AP (2011) Increased methylation variation in epigenetic domains across cancer types. *Nature genetics* 43: 768-775

Reddington JP, Perricone SM, Nestor CE, Reichmann J, Youngson NA, Suzuki M, Reinhardt D, Dunican DS, Prendergast JG, Mjoseng H, Ramsahoye BH, Whitelaw E, Greally JM, Adams IR, Bickmore WA, Meehan RR (2013) Redistribution of H3K27me3 upon DNA hypomethylation results in de-repression of Polycomb target genes. *Genome Biol* 14: R25

2nd Editorial Decision

08 February 2016

Thank you again for submitting your work to Molecular Systems Biology. We are now globally satisfied with the modification made and we will be able to accept your paper for publication in Molecular Systems Biology pending the following minor formatting issues:

- please complete the author checklist (<<http://msb.embopress.org/sites/default/files/additional-assets/EMBO%20Press%20Author%20Checklist%20-MSB.xlsx>>. This file will be published alongside your paper.
- for the HTML version of your paper we would need the following items:
 1. three to four 'bullet points' highlighting the main findings of your study
 2. a short 'blurb' text summarizing in two sentences the study (max. 250 characters)
 3. a 'thumbnail image' (exact width=550 pixels x maximal height=450 pixels, Illustrator, PowerPoint, OmniGraffle or jpeg format), which can be used as 'visual title' for the synopsis section of your paper.
- please rename the file "Bauer et al. Expanded View" into "Appendix".
- If you wish, you can re-integrate the whole materials & Methods section in the main text. Since we are online only, it is not really limited in length. If you prefer to keep some Materials & Methods description in the Appendix, it should be called out "Appendix Materials Methods".
- Figures or tables that are included in the Appendix should be called out as Appendix Fig S1, Appendix Fig S1, Appendix Table S1, Appendix Table S2, etc..(<<http://msb.embopress.org/authorguide#expandedview>>). Please make sure that the numbering used in the main text corresponds to the numbering used in the Appendix.
- Expanded View Tables provided as Excel tables should be numbered EV1, EV2, EV3 and EV4 and called out accordingly.
- Please remove coloured track changes from the Appendix file.

2nd Revision - authors' response

25 February 2016

We are pleased to hear that the revised version of our manuscript entitled “Environment-induced epigenetic reprogramming in genomic regulatory elements in smoking mothers and their children” is now acceptable for publication in *Molecular Systems Biology*.

According to your request we made the necessary formatting changes: renaming the file “Bauer et al. Expanded View” into “Appendix” and removing the colored track changes from the Appendix file. All tables of the Appendix are now Expanded View Tables and are provided as

Excel tables. For the HTML version of our paper we highlighted the major findings of our study in some bullet points and in a two-sentence summary. In addition we added a cartoon visualizing the title of our manuscript.

We would like to thank you again for taking the time and effort to carefully review our paper, which helped us to considerably improve our manuscript.

Corresponding Author Name: Irina Lehmann / Roland Eils
Manuscript Number: MSB-15-6520R